# The prion-like domain of *Drosophila* Imp promotes axonal transport of RNP granules in vivo

Jeshlee Vijayakumar[1], Charlène Perrois[1], Marjorie Heim [1], Luc Bousset[2], Simon Alberti [3,4] & Florence Besse [1]

Prion-like domains (PLDs), defined by their low sequence complexity and intrinsic disorder, are present in hundreds of human proteins. Although gain-of-function mutations in the PLDs of neuronal RNA-binding proteins have been linked to neurodegenerative disease progression, the physiological role of PLDs and their range of molecular functions are still largely unknown. Here, we show that the PLD of *Drosophila* Imp, a conserved component of neuronal ribonucleoprotein (RNP) granules, is essential for the developmentally-controlled localization of Imp RNP granules to axons and regulates in vivo axonal remodeling. Furthermore, we demonstrate that Imp PLD restricts, rather than promotes, granule assembly, revealing a novel modulatory function for PLDs in RNP granule homeostasis. Swapping the position of Imp PLD compromises RNP granule dynamic assembly but not transport, suggesting that these two functions are uncoupled. Together, our study uncovers a physiological function for PLDs in the spatio-temporal control of neuronal RNP assemblies.

---

[1] University Côte d'Azur, CNRS, Inserm, iBV, Nice 06100, France. [2] Paris-Saclay Institute of Neuroscience, Orsay 91505, France. [3] Max Planck Institute of Molecular Cell Biology and Genetics, Dresden 01307, Germany. [4] Center for Molecular and Cellular Bioengineering (CMCB), Biotechnology Center, Technische Universität Dresden, Dresden 01307, Germany. Correspondence and requests for materials should be addressed to F.B. (email: besse@unice.fr)

Following transcription, splicing, and nuclear processing, eukaryotic mRNAs are exported to the cell cytoplasm as ribonucleoprotein (RNP) complexes containing RNA molecules and associated regulatory proteins. Individual RNP complexes can further assemble into higher order structures detected by light microscopy and referred to as RNP granules[1]. Cytoplasmic RNP granules of different sizes, composition, and properties have been defined over the last past decades, including large macromolecular complexes such as P-bodies, stress granules, germ cell granules or neuronal granules[2]. These assemblies are enriched in helicases, regulators of mRNA translation and stability, and/or molecular motors. They represent a very efficient and flexible means to compartmentalize and regulate gene expression[2–4]. Neuronal granules, in particular, have been implicated in the long-distance transport of mRNAs to axons or dendrites, and in their local translation in response to external cues[5–8]. By enabling precise and dynamic spatio-temporal expression of mRNAs involved in cytoskeletal remodeling, synaptic activity, or cell signaling, neuronal granules promote functional and structural plasticity in both developing and mature neurons. Their function underlies many fundamental neuronal processes regulated by extrinsic signals, such as synaptic plasticity, axon or dendrite growth and branching, as well as axon survival or regeneration[6,9–11]. To date, however, the cellular and molecular principles underlying the assembly, transport and regulation of neuronal RNP granules are still poorly understood.

Numerous recent studies have suggested that the assembly of macromolecular RNP granules is mediated by the process of liquid–liquid phase separation, i.e., the demixing of a homogenous solution into a soluble phase in which RNA and associated proteins are dispersed, and a condensed phase in which these components are concentrated in droplets with semi-liquid behavior[12–14]. Once assembled, these RNP droplets undergo constant changes at the molecular level, as illustrated by the relatively fast exchange (from seconds to minutes) of RNP components[15–19]. This property enables RNP granules to rapidly change their size, number, and/or composition, and can be modulated in response to physiological cues or environmental stresses[18,20–23]. As revealed by recent in vitro studies, the establishment of multivalent protein–protein and protein–RNA interactions is a key factor driving the coalescence and maintenance of dynamic RNP assemblies[24–28]. In this context, the role of intrinsically disordered prion-like domains (PLDs), found at high frequency in RNP granule components[29–31], has raised strong interest. PLDs are composed of repeated stretches of uncharged polar and aromatic amino acids, rendering them very interactive and able to drive the formation of transient interaction networks underlying condensation reactions. Consistent with this idea, most PLDs studied so far promote self-assembly in reconstituted systems, and seed phase separation into RNP droplets[19,25,27,29,32]. Interestingly, alterations in PLD functionality have been linked to the progression of several neurodegenerative diseases including amyotrophic lateral sclerosis (ALS) or frontotemporal dementia (FTD)[31,33]. Disease-causing mutations identified in the PLDs of different RNA-binding proteins, indeed, were shown to alter granule properties and promote the formation of abnormal solid aggregates, a hallmark of ALS and FTD[19,34,35]. Surprisingly, although a clear link has now been established between alteration of PLD function and disease, the physiological function of PLDs in the assembly and regulation of neuronal RNP granules largely remains to be demonstrated.

Here, we have explored the role of a PLD found in Drosophila Imp, a known component of neuronal RNP granules belonging to the conserved family of VICKZ RNA-binding proteins. In both vertebrate and invertebrate neurons, Imp family members are packaged together with target mRNAs into microscopically visible granules that are transported to the axons and/or dendrites of neuronal cells[36–39]. As best described for Vg1RBP and ZBP1, two vertebrate orthologs of Imp, Imp proteins not only promote the microtubule-dependent transport of their targets such as β-actin mRNA, but also control their translational regulation[22,40–43]. Functionally, Vg1RBP/ZBP1-dependent localization and translational control of β-actin promote axon navigation[36,42,44], as well as dendritic growth and branching[45] in developing neurons. In Drosophila brains, Imp assembles into neuronal RNP granules that contain the actin regulator-encoding profilin mRNA. These granules undergo precise spatio-temporal regulation during nervous system maturation, and are dynamically recruited to axons upon developmental remodeling[37]. Moreover, Imp function is required for completion of axonal branch remodeling, in particular for the regrowth and branching of adult axons that occur after pruning of immature branches.

In this study, we find that Drosophila Imp PLD promotes the motility of axonal Imp granules in vivo, and is both necessary and sufficient for efficient localization of Imp to axons. Furthermore, we functionally show that Imp PLD regulates the imp-dependent axonal remodeling that occurs during brain maturation. In contrast to other PLD domains, Imp PLD is not required for RNP granule assembly in cells and in vivo. Imp PLD, rather, regulates granule properties by limiting the clustering of Imp molecules, and by promoting their exchange in and out of granules. Such functions do not depend on PLD primary sequence. Strikingly, swapping the position of Imp PLD revealed that its function in the modulation of Imp granule assembly and dynamics can be uncoupled from that in axonal transport. Together, our findings reveal a novel in vivo function for a PLD in the formation of transport-competent neuronal granules. By uncovering an unexpected function of a PLD in RNP granule homeostasis, they also shed new light into the molecular principles and requirements underlying RNP granule assembly and regulation in living cells.

## Results

**Drosophila Imp contains a C-terminal prion-like domain.** Analysis of the primary sequence of Drosophila Imp revealed that the C-terminal most region of the protein is highly enriched in uncharged polar amino acids, in particular in glutamines that are clustered in stretches of 3–5 residues (31 glutamines over a 95 amino acid length; Fig. 1a). This C-terminal domain has an amino acid composition typical of that described for low-complexity prion-like domains (PLD)[46], and was identified as such in a genome-wide in silico search[30]. This domain is further predicted to be intrinsically disordered (Fig. 1b), a prediction that we validated using circular dichroïsm spectroscopy. As shown in Fig. 1c, indeed, spectra obtained from recombinant Imp PLD revealed that this domain does not significantly fold into detectable α-helices or β-sheets secondary structures.

**Imp PLD is important for imp-dependent axonal remodeling.** To investigate the biological function of Imp PLD, we generated flies that express proteins lacking the PLD (Imp-ΔPLD) using the CRISPR/Cas9 technology. Specifically, we introduced premature STOP codons upstream of the PLD-coding sequence, in the imp locus of an available knock-in line containing a GFP exon inserted in frame in all imp transcripts (Supplementary Fig. 1a, b). In this original G080 protein-trap line, functional endogenously tagged GFP-Imp proteins are produced[37]. Strikingly, G080-GFP-Imp-CRISPR-ΔPLD homozygous flies, in contrast to flies lacking the function of imp[47], were homozygous viable and fertile, suggesting that Imp PLD is not required to support the essential functions of the Imp protein. Beyond having an essential (but not

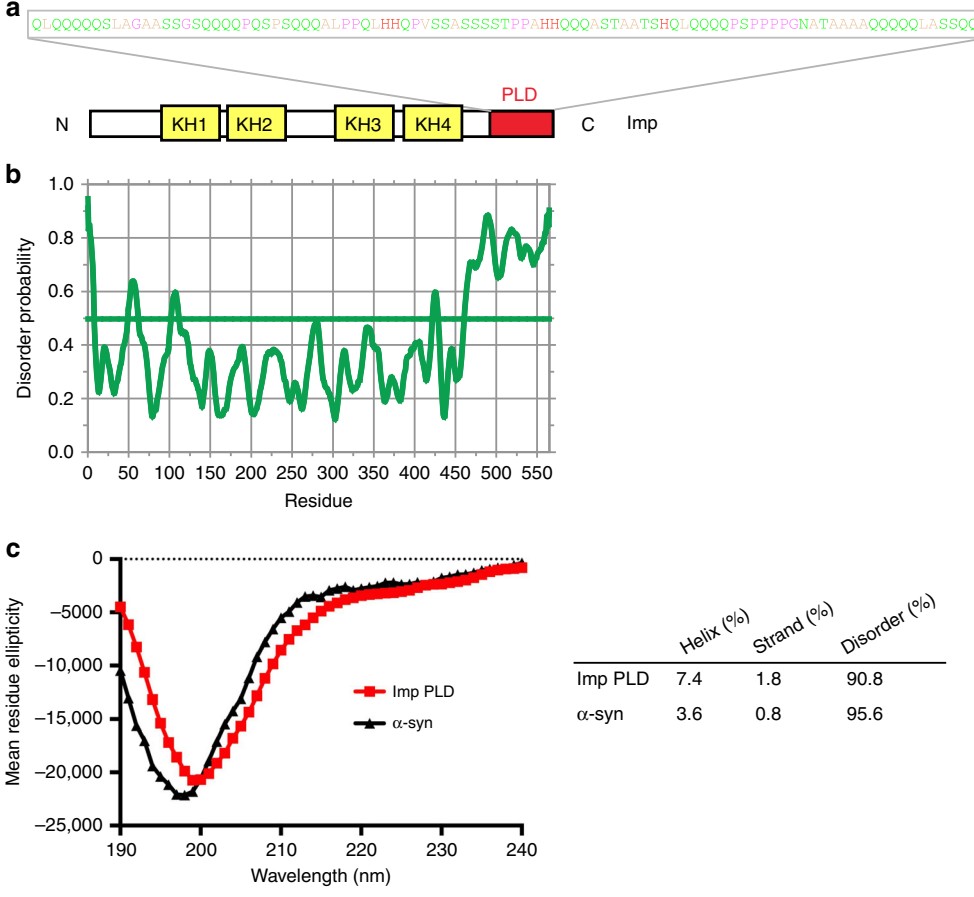

**Fig. 1** *Drosophila* Imp contains a C-terminal prion-like domain (PLD). **a** Schematic representation of *Drosophila* Imp (PB isoform). The four KH RNA-binding domains are shown in yellow, and the prion-like domain (PLD) in red. Residues composing the prion-like domain are colored according to their physicochemical properties (Zappo color code). **b** Plot of the degree of disorder along the Imp protein, as predicted by the DisEMBL Intrinsic Protein Disorder Prediction 1.5 algorithm. **c** Far ultraviolet circular dichroïsm spectrum of the PLD of Imp (red). The spectrum of α-Synuclein (α-Syn, black) is shown for comparison. Predicted fractions of structured and unstructured regions are shown in the table for both peptides. Source data are provided as a Source Data file

characterized) function at metamorphosis onset, *imp* was shown to be required for the developmentally controlled remodeling of Mushroom Body (MB) γ neurons (Fig. 2a) in the maturing brain. Imp, while dispensable for the initial growth of larval MB γ axonal branches, is required for the growth and branching of adult γ axonal processes that occur during metamorphosis, subsequently to the pruning of larval immature branches[37,48]. As shown in Fig. 2b, c, G080-GFP-Imp-CRISPR-ΔPLD flies did not exhibit major alterations in the global projection pattern of the adult γ axon population. To visualize the detailed morphology of individual γ neurons in this line, we induced stochastic sparse labeling of γ neurons using the MultiColor FlpOut (MCFO) approach[49]. While the majority of single labeled adult γ axons had a normal morphology, about 10% of them exhibited polarized growth defects (6/62 *vs* 0/21 in control flies) (Fig. 2d, e). In this context, all γ neurons expressed truncated Imp-ΔPLD proteins. To then test the function of Imp-ΔPLD proteins in a condition where single mutant neurons are challenged by surrounding wild-type neurons growing simultaneously and competing for space[50], we performed rescue experiments using the MARCM technique[51]. About half of individual *imp*7 mutant neurons thus generated in an otherwise wild-type environment failed to properly elongate, a phenotype significantly suppressed by expression of wild-type GFP-Imp (Fig. 2f–h). Remarkably, expression of GFP-Imp-ΔPLD did not suppress the axon growth defects observed in adult *imp* mutant γ neurons in this assay

(Fig. 2h), indicating that the PLD of Imp is important for efficient axon regrowth in vivo, a process better highlighted in a competitive context.

**Imp PLD promotes the transport of Imp to MB γ axons**. We have previously shown that the function of *imp* in developmental axon regrowth temporally correlates with its recruitment to axons in vivo[37]: while Imp is restricted to the cell bodies of MB γ neurons during larval stages, it localizes to axons from early metamorphosis onwards. To investigate the role of Imp PLD in the recruitment of Imp to axons, we first analyzed at adult stages the localization of GFP-Imp fusions expressed in MB γ neurons under the control of 201Y-Gal4. In contrast to GFP-Imp, which localized throughout the adult MB γ lobes in a granular pattern (Fig. 3a, c, d), GFP-Imp-ΔPLD proteins did not efficiently localize to the axons of adult γ axons (Fig. 3b, c), suggesting that Imp PLD is required in vivo to promote the localization of Imp to axons. To assess the distribution of endogenously-expressed Imp proteins, we then compared the localization of GFP-Imp proteins produced from the G080 protein-trap line with that of GFP-Imp-ΔPLD proteins produced from the G080-GFP-Imp-CRISPR-ΔPLD line. As shown in Fig. 3f–h, a strong reduction in the accumulation of GFP-Imp proteins in MB γ axons was observed in the absence of Imp PLD. Such a difference in axonal signal intensity reflected a decreased density of Imp-containing axonal

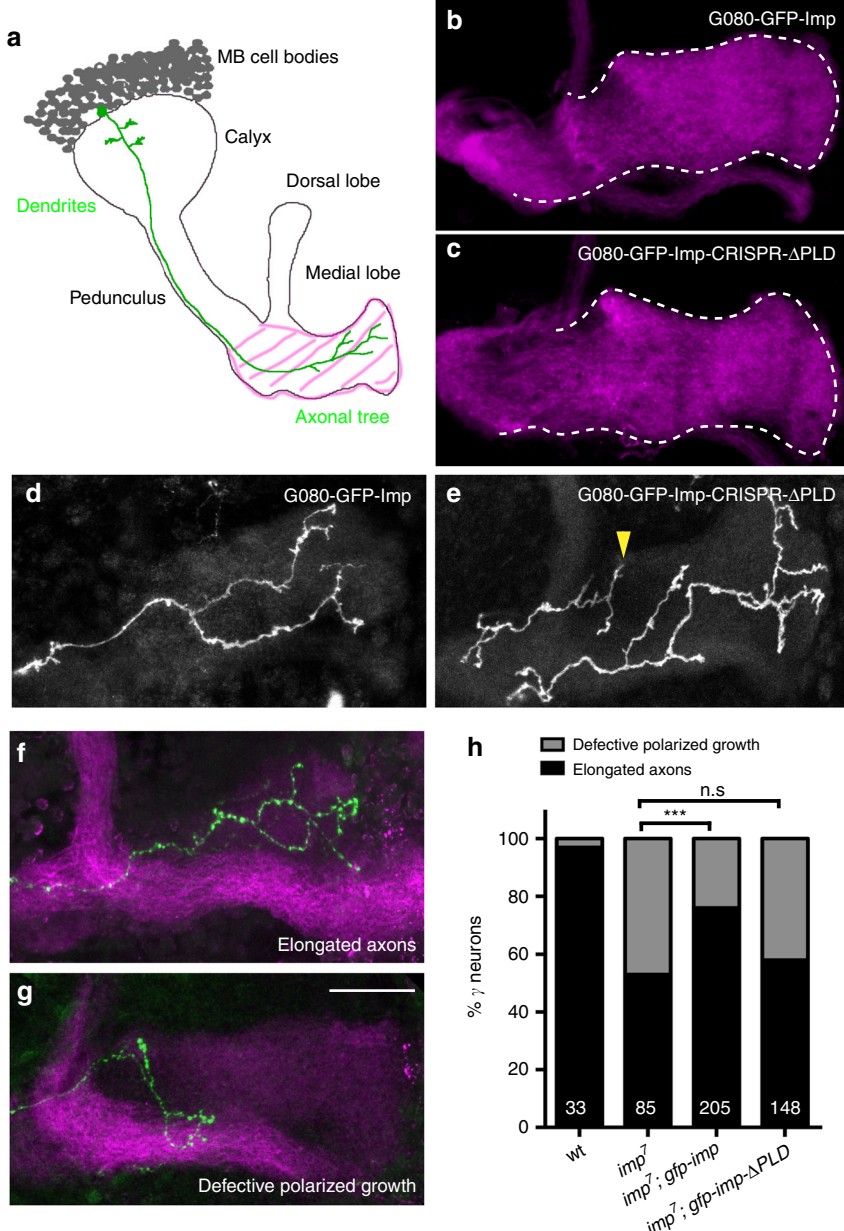

**Fig. 2** Imp PLD is required for in vivo axonal remodeling. **a** Schematic representation of adult Mushroom Body (MB) structure. The medial lobe, which corresponds to the distal part of the γ axon population and is stained in magenta in **b**, **c**, **f**, **g**, is highlighted with pink stripes. The morphology of a single γ neuron is represented in green. Adapted with permission from[37]. **b**, **c** Medial lobes of adult G080-GFP-Imp (**b**) and G080-GFP-Imp-CRISPR-ΔPLD (**c**) MBs labeled with 201Y-Gal4, UAS-RFP. The RFP signal is shown in magenta. **d**, **e** Individual γ axons from G080-GFP-Imp (**d**) and G080-GFP-Imp-CRISPR-ΔPLD (**e**) homozygous adult brains labeled with the MCFO technique. The yellow arrowhead points to a not properly elongated γ axon. **f**, **g** Representative images of adult MB γ neurons with properly elongated axons (**f**), or showing defective polarized growth (**g**). Single mutant axons were generated and labeled by GFP (green) using the MARCM technique. The medial lobe is stained with FasciclinII (magenta). Scale bar in **b**-**g**: 20 μm. **h** Percentages of adult γ axons that succeeded (elongated axon) or failed (defective axonal growth) to reach the extremity of the medial lobe. ***P < 0.001 (Fisher's exact test). n. s. stands for not significant. Numbers correspond to the total numbers of scored individual axons. Source data are provided as a Source Data file. Complete genotypes: FRT19A, tub-Gal80, hsp-flp/FRT19A; 201Y-Gal4, UAS-GFP/+ (wt); FRT19A, tub-Gal80, hsp-flp/FRT19A imp[7]; 201Y-Gal4, UAS-GFP/+ (imp mutant) and FRT19A, tub-Gal80, hsp-flp/FRT19A imp[7]; 201Y-Gal4, UAS-GFP/UAS-gfp-imp or UAS-gfp-imp-ΔPLD (rescues)

puncta (Fig. 3i, j). Furthermore, it was not explained by differences in expression levels, as protein levels were similar in the G080-GFP-Imp and G080-GFP-Imp-CRISPR-ΔPLD lines (Supplementary Figs. 1b and 2a).

To test whether Imp PLD is sufficient to promote the axonal recruitment of heterologous proteins, we generated a chimeric construct in which we grafted the PLD of *Drosophila* Imp at the C-terminus of the human IMP1 protein, which naturally lacks

such a domain (Supplementary Fig. 2b). Remarkably, a significant increase in axonal localization was observed upon addition of *Drosophila* Imp PLD to hIMP1 (Fig. 3k–m), indicating that Imp PLD is both necessary and sufficient for axonal localization.

**Imp PLD molecular properties underlying axonal localization.**
To better understand the molecular determinants underlying Imp PLD function in axonal localization, we analyzed the function of

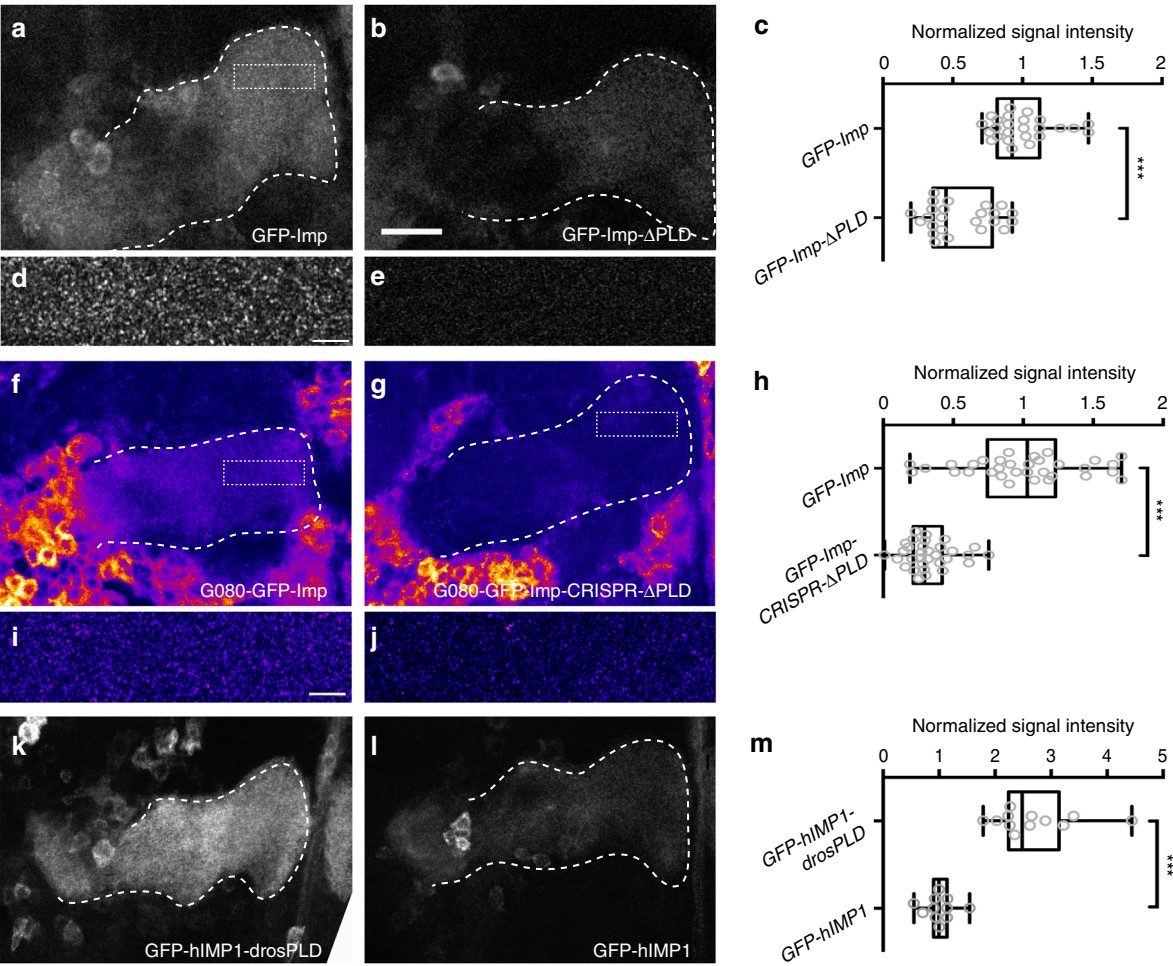

**Fig. 3** Imp PLD promotes efficient localization to axons in vivo. **a**, **b** Medial lobes of adult brains expressing GFP-Imp (**a**) or GFP-Imp-ΔPLD (**b**) under the control of the 201Y-Gal4 driver. **c** Distributions of normalized GFP signal intensities in distal axons. **d**, **e** Magnified views of the distal region of medial γ lobes expressing GFP-Imp (**d**) or GFP-Imp-KH1–4DD (**e**), a form that cannot bind RNA and does not form granules (see Fig. 5). GFP-Imp-KH1–4DD is used to show the specificity of the granular pattern. Note that the image shown in **d** was not taken from the brain shown in **a**. Still, the approximate position of the magnified region is boxed in **a**. **f**, **g** Medial lobes of adult brains homozygous for the G080-GFP-Imp (**f**) or G080-GFP-Imp-CRISPR-ΔPLD (**g**) chromosomes. Note that the cell bodies of surrounding cells also expressing Imp are visible in **f**, **g**. GFP signals are shown with the Fire look-up table of ImageJ. **h** Distributions of normalized GFP signal intensities in distal axons. **i**, **j** Magnified views of the distal region of G080-GFP-Imp (**i**) or G080-GFP-Imp-CRISPR-ΔPLD (**j**) medial γ lobes. Note that images in **i**, **j** were not taken from the brains shown in **f**, **g**. Still, the approximate position of the magnified regions is boxed in **f**, **g**. Distributions of normalized GFP signal intensities in distal axons. **k**, **l** Medial lobes of adult brains expressing GFP-hIMP1-drosPLD (**k**) or GFP-hIMP1 (**l**) under the control of the 201Y-Gal4 driver. **m** Distributions of normalized GFP signal intensities in distal axons. The dashed lines in **a**–**l** delimit the distal part of MB γ axon bundles (medial lobes; see Fig. 2a). Genotypes: 201Y-Gal4/UAS-GFP-Imp (**a**, **d**); 201Y-Gal4/UAS-GFP-Imp-ΔPLD (**b**); 201Y-Gal4/UAS-GFP-ImpKH1–4DD (**e**); 201Y-Gal4/UAS-GFP-hIMP1-drosPLD (**k**) and 201Y-Gal4/UAS-GFP-hIMP1 (**l**). All box plots are represented using the min to max convention, where the middle line defines the median and the whiskers go down to the smallest value and up to the largest. Numbers of MB analyzed: UAS-GFP-Imp: 25; UAS-GFP-Imp-ΔPLD: 24; G080-GFP-Imp protein-trap: 33; G080-GFP-Imp-CRISPR−ΔPLD: 32; UAS-GFP-hIMP1: 11; UAS-GFP-hIMP1-drosPLD: 12. *** $P < 0.001$ (Mann–Whitney test). Scale bars in **a**, **b**, **f**, **g**, **k**, **l**: 15 µm. Scale bars in **d**, **e**, **i**, **j**: 3 µm. Source data are provided as a Source Data file

variants in which Imp PLD sequence or position was altered. First, we tested whether Imp PLD might encode interaction sites with specific sequence by generating two scrambled PLD variants with altered primary sequence but preserved overall amino acid composition (Imp-scr2 and Imp-scr4; Fig. 4a and Supplementary Fig. 2b). In these two PLD variants, the degree of disorder is predicted to be preserved (Supplementary Fig. 2c). Remarkably, GFP-Imp-scr2 and GFP-Imp-scr4 proteins expressed in MB γ neurons localized to axons similarly to wild-type proteins (Fig. 4b–d, f), indicating that primary sequence is not important for Imp PLD function in axonal localization, and thus that Imp PLD does not regulate this process through stereospecific interactions. To next test whether adding the PLD N-terminally could

compensate for the lack of C-ter PLD, we generated a form of Imp in which we moved the PLD from the C-terminus of the protein to its N-terminus (Supplementary Fig. 2b). Expression of GFP-Imp-Nter-PLD in MB γ neurons revealed that this variant is able to localize to axons as efficiently as wild-type GFP-Imp fusions (Fig. 4e, g), suggesting that the presence of Imp PLD, but not its context, is important for axonal transport.

**Imp PLD is not required for Imp granule assembly.** As previously described, Imp is transported to axons as granules undergoing active motion[37]. Given that PLDs were shown in different contexts to drive the coalescence of RNP

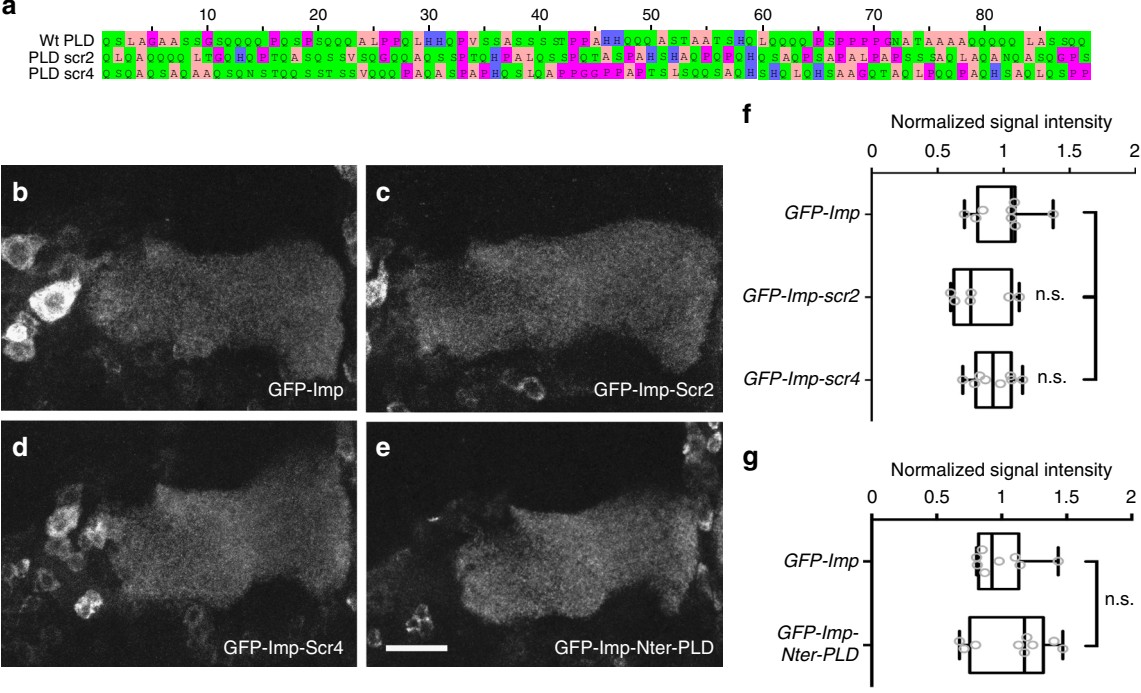

**Fig. 4** Molecular requirements of PLD function in axonal localization. **a** Primary sequences of the wild-type (top) and scrambled (bottom) PLDs. Scramble variants were generated randomly and exhibit different degree of Glutamine dispersion. **b–e** Medial lobes of adult brains expressing GFP-Imp (**b**), GFP-Imp-Scr2 (**c**), GFP-Imp-Scr4 (**d**), or GFP-Imp-Nter-PLD (**e**) under the control of the 201Y-Gal4 driver. Genotypes: 201Y-Gal4/UAS-GFP-Imp or UAS-GFP-Imp-Scr2 or UAS-GFP-Imp-Scr4 or UAS-GFP-Imp-Nter-PLD. Scale bar in **b–e**: 15 μm. **f**, **g** Distributions of normalized GFP signal intensities in distal axons. Box plots are represented using the min to max convention, where the middle line defines the median and the whiskers go down to the smallest value and up to the largest. Numbers of MB analyzed: UAS-GFP-Imp: 8 (for **f**, **g**); UAS-GFP-Imp-Scr2: 6; UAS-GFP-Imp-Scr4: 8; UAS-GFP-Imp-Nter-PLD:9. ns stands for not significant (Mann–Whitney test). Source data are provided as a Source Data file

granule components into phase-separated higher order structures[19,25,27,29], we investigated the importance of Imp PLD in the assembly of Imp granules in vivo. Both wild-type and ΔPLD GFP-Imp proteins expressed in MB γ neurons via the 201Y-Gal4 driver were found within distinct cytoplasmic granules in cell bodies, similarly to the endogenous protein (Fig. 5a–d). In these experiments, the presence of endogenous wild-type Imp proteins may influence the distribution of exogenous GFP constructs. Thus, we analyzed the distribution of GFP-Imp-ΔPLD proteins produced by homozygous G080-GFP-Imp-CRISPR-ΔPLD flies. Remarkably, GFP-Imp-ΔPLD proteins expressed in this condition also assembled into cytoplasmic granules (Fig. 5e), indicating that the PLD of Imp is not required for granule assembly. RNA was recently shown in vitro and in cells to play a preponderant role in the phase behavior of RNA-binding proteins[26,52]. To test if it is required in vivo for granule assembly, we generated a mutant form of Imp in which two negatively charged aspartate residues were introduced in the characteristic GxxG loop of all four KH domains (GxxG to GDDG substitutions). Such point mutations were described to preserve KH domain structure while strongly impairing nucleic acid binding[53]. As expected, Imp-KH1–4DD proteins did not show any significant binding to RNA in vitro (Fig. 5f). When expressed in MB γ neurons, GFP-Imp-KH1–4DD proteins did not assemble into visible granules (Fig. 5g), suggesting that the binding of Imp to target RNAs, but not Imp PLD, is the main driver of Imp RNP granule assembly.

To determine whether the lack of Imp PLD may alter the composition of Imp RNP granules, we searched for other granule components. As shown in Supplementary Fig. 3a and b, neither wild-type Imp nor Imp-ΔPLD granules did contain stress granule markers (Rin/G3BP, eIF4G, and FMRP). Furthermore, both wild-

type Imp and Imp-ΔPLD granules contained previously described components of neuronal transport granules (Staufen, Me31B, Trailer-hitch, Pur-α, and eIF4e; Supplementary Fig. 3b, c)[8,54,55]. Finally, *profilin mRNA*, a direct and functional mRNA target of Imp[37], was found associated with both wild-type and ΔPLD granules (Supplementary Fig. 3d, e). Together, these results indicate that granule composition is globally preserved in the absence of Imp PLD.

**Imp PLD promotes the motility of Imp RNP granules.** Recruitment of Imp granules to axons is mediated by bidirectional, microtubule-dependent transport, a process triggered during early metamorphosis by a yet unknown instructive signal[37]. To investigate whether Imp PLD regulates the motility of Imp granules in vivo, we performed real-time imaging of endogenous axonal Imp granules in G080-GFP-Imp and G080-GFP-Imp-CRISPR-ΔPLD intact pupal brains[56]. As previously reported[37], wild-type GFP-Imp granules exhibited a biased bidirectional motion (Fig. 6a, b and Supplementary Movie 1) characterized by a higher number of granules moving anterogradely (Fig. 6c and Supplementary Fig. 4a) and a higher anterograde mean velocity (Fig. 6d). Remarkably, a decreased number of both unidirectional and bidirectional motile particles was observed in G080-GFP-Imp-CRISPR-ΔPLD axons compared to wild-type G080-GFP-Imp axons (Fig. 6c; $P < 0.01$ in a Kruskal–Wallis test). Furthermore, although the proportion of anterograde particles was still higher than that of retrograde ones in G080-GFP-Imp-CRISPR-ΔPLD brains (Supplementary Fig. 4a), an increased retrograde mean velocity was observed (Fig. 6d and Supplementary Fig. 4b,c). Together, these results thus suggest that the PLD of Imp promotes the motility of

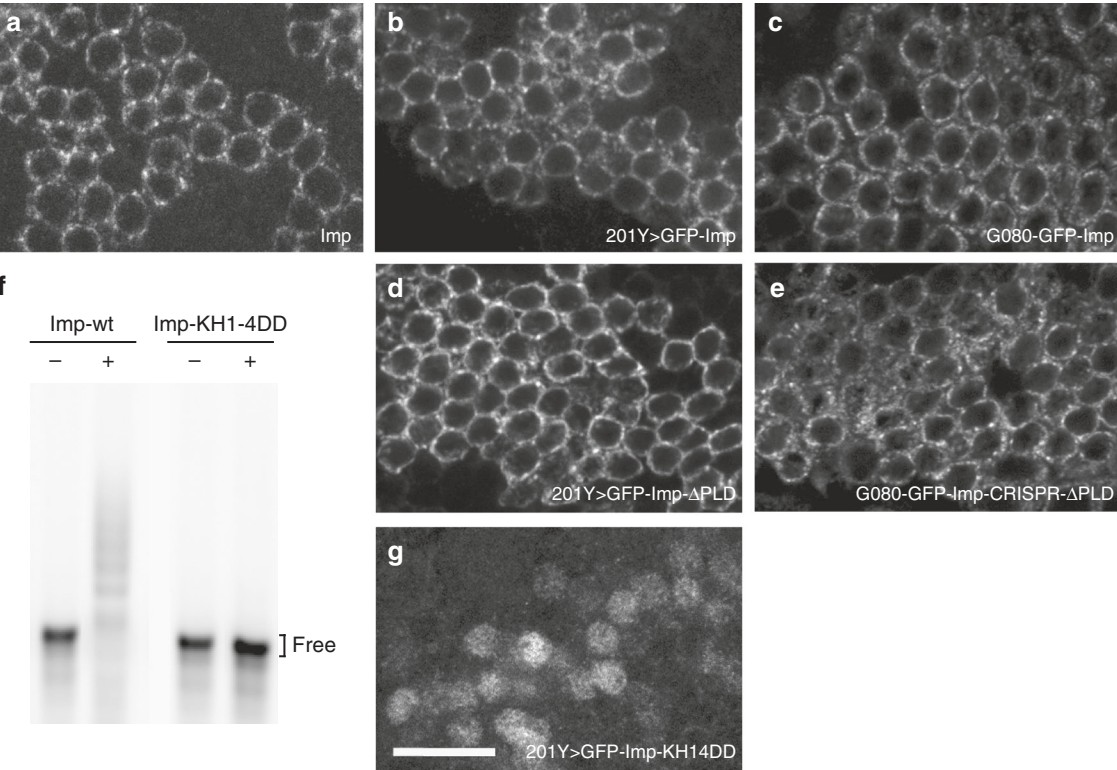

**Fig. 5** Imp PLD is dispensable for RNP granule assembly. **a** Cell bodies of wild-type adult MB γ neurons stained with anti-Imp antibodies. **b, d, g** Cell bodies of adult MB γ neurons expressing wild-type GFP-Imp (**b**), GFP-Imp-ΔPLD (**d**) or GFP-Imp-KH1–4DD (**g**) under the control of the 201Y-Gal4 driver. Complete genotypes: 201Y-Gal4/UAS-GFP-Imp, 201Y-Gal4/UAS-GFP-Imp-KH1–4DD, and 201Y-Gal4/UAS-GFP-Imp-ΔPLD. **c, e** Cell bodies of adult MB γ neurons homozygous for the G080-GFP-Imp (**c**) or G080-GFP-Imp-CRISPR−ΔPLD (**e**) chromosomes. GFP signals are shown in white. Note that the KH1–4DD mutations induce an accumulation of Imp proteins in the nucleus, as described for the vertebrate protein[68]. Scale bar in **a–g**: 10 μm. **f** EMSA analysis using fluorescently-labeled *profilin* 3′UTR in the absence (−) or presence (+) of 800 nM recombinant GFP-Imp (left) or GFP-Imp-KH1–4DD (right)

endogenous axonal Imp granules in vivo and modulates the properties of their retrograde transport.

**Imp PLD modulates granule size and number**. To test whether the observed changes in granule motility may be linked to changes in the material properties of Imp RNP granules, we carefully quantified the propensity of wild-type and mutant Imp molecules to condensate into visible granules using cultured S2R+ cells. In these cells, both endogenous Imp and transfected GFP-Imp distributed diffusely throughout the cytoplasm and in addition accumulate within punctate structures of an apparent diameter of 200–300 nm (Fig. 7a, b). These granules were observed in the absence of stress, and did not colocalize with stress granule markers (Supplementary Fig. 5a,b). Careful quantification of the cytoplasmic granules formed upon expression of GFP-tagged proteins revealed that, although GFP-Imp and GFP-Imp-ΔPLD constructs were expressed at similar levels (Supplementary Fig. 6a), a significantly higher number of granules was observed in GFP-Imp-ΔPLD-expressing cells compared to GFP-Imp-expressing cells (Fig. 7c, d and Supplementary Fig. 6b). GFP-Imp-ΔPLD granules exhibited a range of sizes (Supplementary Fig. 6c), but frequently reached abnormally high values (Fig. 7e). These results suggest that Imp PLD normally restricts the number and size of Imp granules. Next, we wondered whether Imp PLD is sufficient to limit the assembly of RNP granules. On its own, Imp PLD does not trigger particle assembly and cannot be recruited to Imp-containing granules (Supplementary Fig. 6d). Strikingly, addition of *Drosophila* Imp PLD to human IMP1 significantly decreased both the number and the size of cytoplasmic granules formed by hIMP1 in S2R+ cells (Fig. 7f–i and Supplementary Fig. 6f), without affecting hIMP1 levels

(Supplementary Fig. 6e). Together, these results thus indicate that Imp PLD is not only required, but also sufficient, to modulate the transitioning of Imp molecules into granules.

Notably, PLD primary sequence is not important for the regulation of Imp molecule coalescence, as Imp-scr2 and Imp-scr4 assembled into granules similar to wild-type ones (Supplementary Fig. 7a–e), and grafting scrambled PLDs onto hIMP1 restricted granule assembly as efficiently as wild-type PLD (Supplementary Fig. 7f). Imp PLD, however, encodes specific information, as its absence could not be compensated by an irrelevant sequence of similar length (truncated GFP sequence, see Supplementary Fig. 7g).

**Imp PLD renders Imp RNP granules more dynamic**. RNP granules are dynamic assemblies whose components constantly exchange with the cytoplasm. To test if Imp PLD regulates the exchange of granule-associated Imp, we measured fluorescence recovery after photobleaching (FRAP) in GFP-Imp-expressing S2R+ cells. As shown in Fig. 8a, b, a partial recovery was observed after photobleaching of the granule-associated pool of wild-type GFP-Imp proteins, suggesting that Imp may be a core particle component that moderately exchanges with the cytoplasmic fraction. Interestingly, a significantly lower recovery was observed when expressing GFP-Imp-ΔPLD fusions (Fig. 8b). Such a reduction is not linked to differences in granule size, as a decreased fluorescence recovery was observed for both small and large granules in the absence of Imp PLD (Supplementary Fig. 8a). Similar results were also obtained when expressing GFP-Imp-ΔPLD constructs in HeLa cells, a heterologous system where the identity of Imp partners and targets is likely partially different

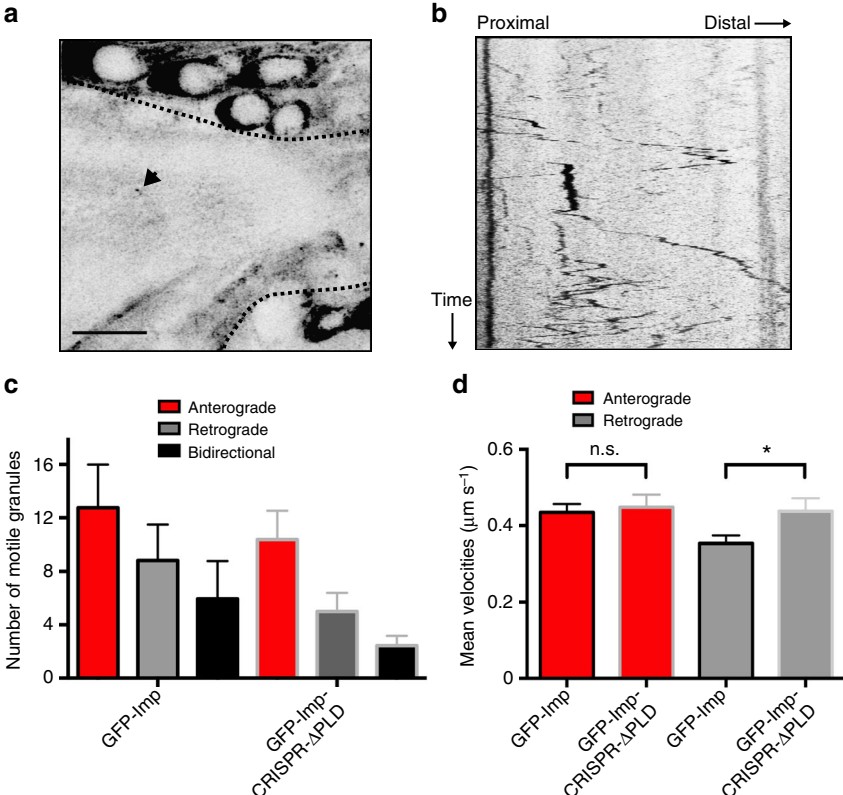

**Fig. 6** Imp PLD promotes Imp granule motility. **a**, **b** Single image (**a**) and kymograph (**b**) extracted from a video generated from a G080-GFP-Imp hemizygous pupal brain (24 h APF; see also Supplementary Movie 1). The bundle of MB γ axons is delimited by dotted black lines. The arrowhead points to a GFP-Imp granule. Scale bar: 10 μm. **c** Average number of motile anterograde, retrograde and bidirectional GFP-Imp granules per 12 min-long movie. Numbers of movies analyzed: $n = 17$ for G080-GFP-Imp and 16 for GFP-Imp-CRISPR-ΔPLD. Total numbers of granules analyzed: 468 for G080-GFP-Imp and 284 for G080-GFP-Imp-CRISPR-ΔPLD. **d** Mean anterograde and retrograde velocities. Numbers of anterograde granules analyzed: 271 (G080-GFP-Imp) and 178 (G080-GFP-Imp-CRISPR-ΔPLD); numbers of retrograde granules analyzed: 192 (G080-GFP-Imp) and 91 (G080-GFP-Imp-CRISPR-ΔPLD). Error bars in **c**, **d** represent s.e.m. *$P < 0.05$ (Unpaired $t$-test). ns stands for not significant. Source data are provided as a Source Data file

(Supplementary Fig. 8b). Furthermore, decreased recovery was also visible in whole adult brains, after bleaching of endogenous GFP-Imp and GFP-Imp-ΔPLD neuronal granules (Fig. 8c, d). Notably, fluorescence recovery of GFP-Imp-scr2 and GFP-Imp-scr4 variants was similar to that of wild-type GFP-Imp proteins (Fig. 8e), consistent with a model in which granule dynamics is not regulated through PLD-mediated stereospecific interactions.

Altogether, these results suggest that Imp PLD promotes the exchange of Imp molecules in and out the granules. Instead of promoting granule assembly, it plays an important role in modulating RNP granule assembly and dynamics.

**Defects in granule properties and transport are uncoupled**. As described previously, Nter-PLD-Imp proteins localized normally to axons (Fig. 4e, g). Furthermore, expressing these proteins in an *imp* mutant background suppressed the axonal regrowth phenotypes observed upon *imp* inactivation (Fig. 9a), suggesting that the presence of an Nter PLD can support in vivo axonal transport and axonal regrowth. Intriguingly, however, GFP-Imp-Nter-PLD variants appeared to be defective in regulating granule material properties. Increased granule number and size were observed upon expression of GFP-Imp-Nter-PLD in S2R+ cells (Fig. 9c–e), and a significantly lower fluorescence recovery was observed in FRAP experiments (Fig. 9b). Thus, these results indicate that Imp PLD has two independent functions in regulating Imp RNP granule assembly and axonal transport. Furthermore, they indicate that changes in RNP granule dynamic properties do not

directly impact on the in vivo function and regulation of Imp during developmental neuronal remodeling.

## Discussion
Establishment of multivalent interactions is driving the self-assembly of components into phase-separated high-order structures[12,24]. In the context of RNP granules, multivalency was proposed to originate from repeats of RNA-binding domains[24], as well as from disordered low-complexity domains, which are prone to establish interaction networks and are found at high frequency in RNA-binding proteins[13,29,30,32]. Here, we showed that preventing Imp binding to RNA by mutating all four KH domains interferes with Imp granule formation, consistent with the idea that RNA binding is a key driver of phase separation[26], and that multivalent protein–RNA interactions may underly granule formation[24]. Imp PLD, however, is neither sufficient nor necessary for such a process. This finding contrasts with the capacity of described PLDs to trigger demixing into phase-separated structures[19,25,27,29,57], but is consistent with recent work showing that low-complexity sequences may not necessarily act as seeds for granule coalescence and rather modulate such a process[21,32,58]. Although the non-functionality of recombinant Imp proteins prevented any analysis of Imp behavior in reconstituted in vitro systems, our results obtained in cells show that Imp PLD restricts the capacity of Imp granule components to self-assemble into granules and reduces the exchange of Imp molecules with the cytoplasm. To our knowledge, Imp PLD is the first PLD that negatively regulates demixing. How this domain is

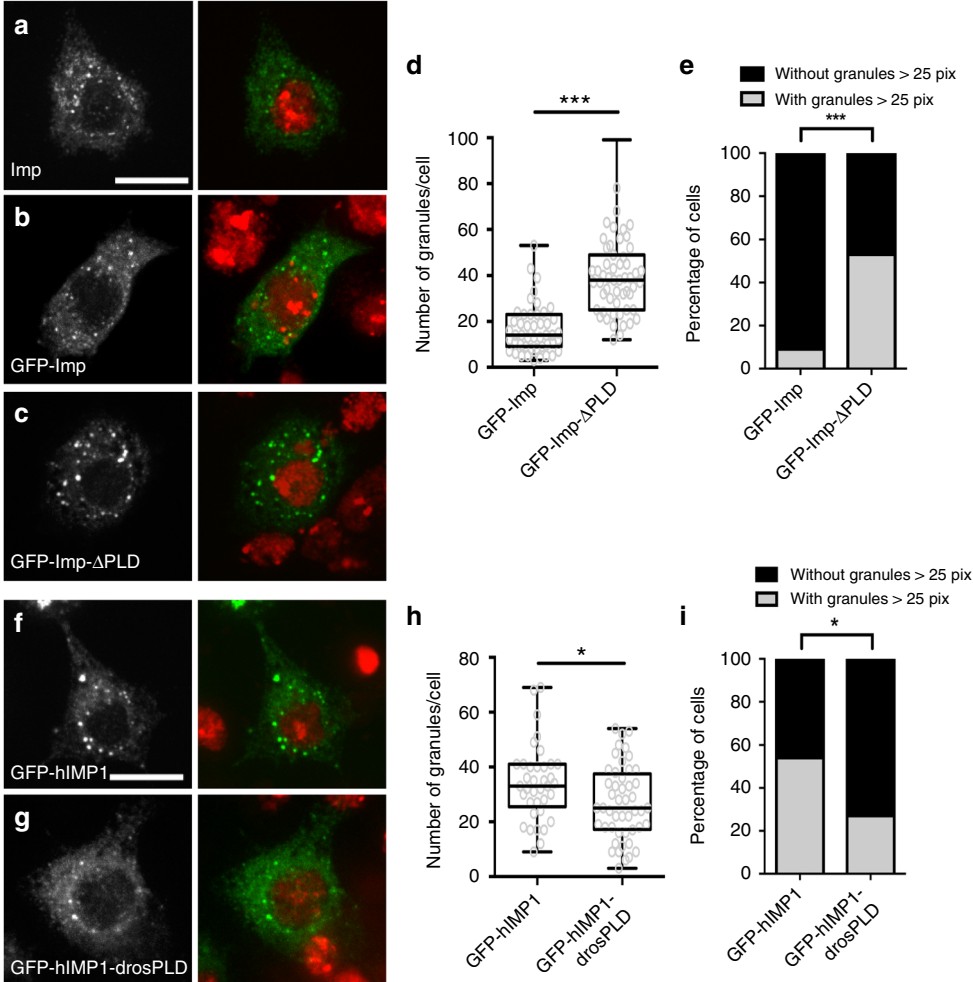

**Fig. 7** Imp PLD modulates granule size and number. **a** S2R+ cell stained with anti-Imp antibodies (left, green in the overlay), and DAPI (red in the overlay). **b, c** S2R+ cells transfected with GFP-Imp (**b**), or GFP-Imp-ΔPLD (**c**) constructs, and stained with DAPI (red in the overlay). GFP signals are shown in white (left) and green in the overlay (right). Scale bar in **a–c**: 10 μm. **d** Distribution of cells in function of their number of GFP-Imp (left) or GFP- Imp-ΔPLD (right) granules. ***$P < 0.001$ (Mann–Whitney test). **e** Percentage of cells exhibiting granules larger than 25 pixels. ***$P < 0.001$ (Fisher's exact test). 47 and 55 cells were analyzed for GFP-Imp and GFP-Imp-ΔPLD constructs respectively. **f, g** S2R+ cells transfected with GFP-hIMP1 (**f**) or GFP-hIMP1-drosPLD (**g**) constructs, and stained with DAPI (red in the overlay). GFP signals are shown in white (left) and green in the overlay. Scale bar: 10 μm. **h** Distribution of cells in function of their number of granules. *$P < 0.05$ (Mann–Whitney test). **i** Percentage of cells exhibiting granules larger than 25 pixels. *$P < 0.05$ (Fisher's exact test). 37 and 49 cells were analyzed for GFP-hIMP1 and GFP-hIMP1-drosPLD, respectively. All box plots are represented using the min to max convention, where the middle line defines the median and the whiskers go down to the smallest value and up to the largest. Source data are provided as a Source Data file

acting at the molecular level still has to be resolved, but one possibility is that it may interfere with intra- and/or inter-molecular interactions, thus decreasing valency of binding or strengths of interactions within the complex. Imp PLD may act in *cis*, by regulating access to Imp binding sites/domains, or in *trans*, by dynamically binding to other granule components, in particular those with intrinsically disordered regions (see Supplementary Fig. 3b). Strikingly, Imp PLD still modulates RNP assembly when ectopically grafted onto a heterologous protein, indicating that its modulatory function is transferable to a related protein. Imp PLD functionality, however, depends on its position within the protein because a N-terminally-located PLD cannot restrict granule assembly. This suggests topological constraints within the interaction network that is established during RNP granule assembly. What is the in vivo impact of changes in granule dynamic properties? Imp PLD modulatory role in granule homeostasis appears to not have a major impact on Imp function during *Drosophila* development, as GFP-Imp-CRISPR-ΔPLD

individuals, in contrast to *imp* mutants, are homozygous viable and fertile. Furthermore, altered granule homeostasis does not interfere with *imp*-dependent remodeling of MB γ axons, as revealed by the capacity of Imp-Nter-PLD proteins to efficiently rescue the *imp* regrowth and branching phenotypes. The capacity of Imp PLD to regulate Imp granule coalescence and dynamics might, however, be physiologically important in other contexts, to regulate synaptic plasticity in the adult nervous system, or in response to stress[31,33].

PLDs belong to a class of low-complexity domains defined by their biased amino acid composition, and in particular by their enrichment in uncharged polar amino acids such as glutamine, asparagine, serine, or proline, a characteristic signature of prion domains[30,46]. A key feature of PLDs is also their predicted lack of defined structure, a property we have experimentally confirmed for Imp PLD using circular dichroïsm spectroscopy. Because of their intrinsic disorder, PLDs have been proposed to bring fuzziness to macromolecular RNP complexes, and to provide both

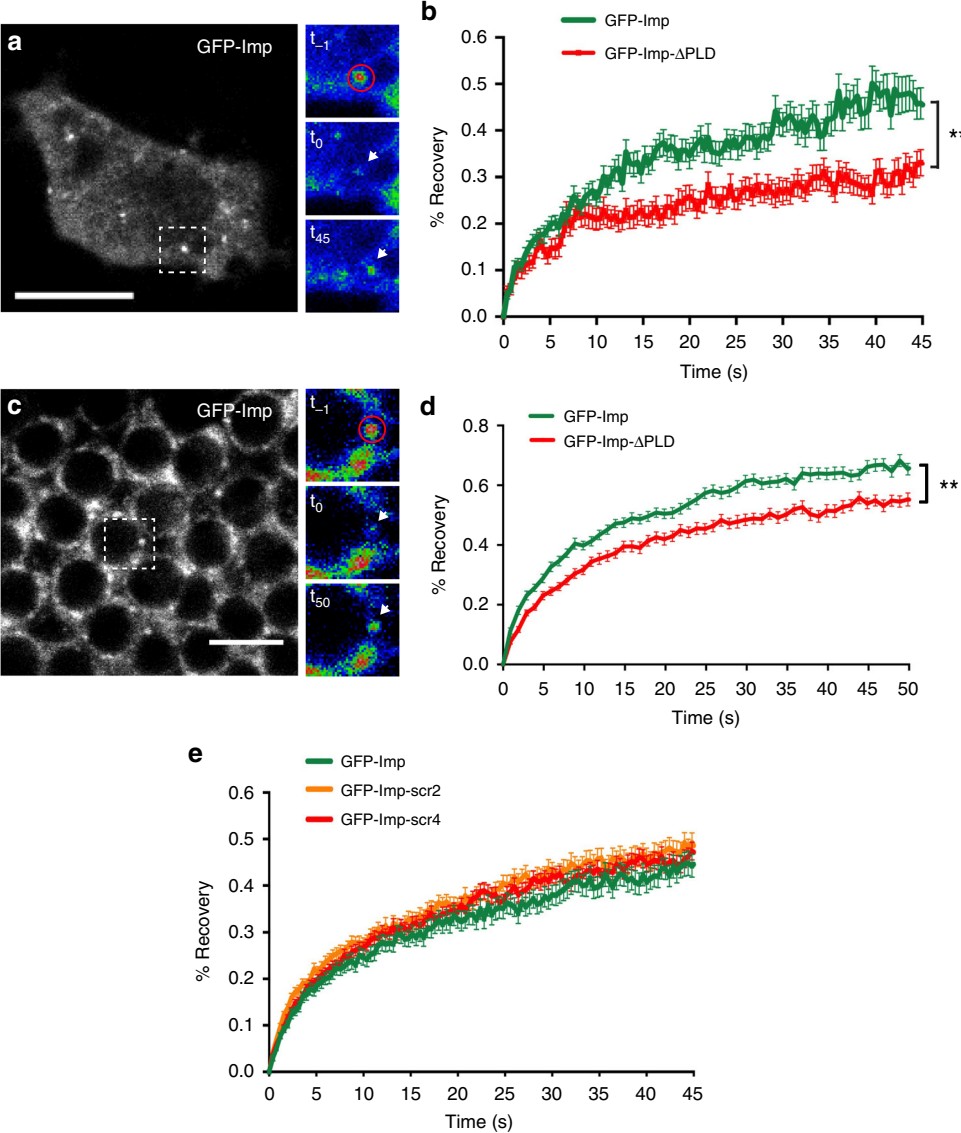

**Fig. 8** Imp PLD promotes Imp granule dynamics. **a** S2R+ cell expressing GFP-Imp. The dashed box indicates the region shown on the right. The red circle delimits the photobleached region, and the white arrows the position of the bleached granule over time. $t_{-1}$ and $t_0$ correspond, respectively, to pre- and post-bleaching time points. $t_{45}$ corresponds to the last recorded time point (t = 45 s). Images were color-coded using the Rainbow RGB function of ImageJ. Scale bar in **a**: 10 μm. **b** Average FRAP curves obtained after photobleaching of GFP-positive particles from S2R+ cells. The following numbers of granules were analyzed: GFP-Imp: 44; GFP-Imp-ΔPLD: 41. **c** Cell bodies of adult MB γ neurons homozygous for the G080-GFP-Imp protein-trap insertion (brain explant). The dashed box indicates the region shown on the right. The red circle delimits the photobleached region, and the white arrows the position of the bleached granule over time. $t_{-1}$ and $t_0$ correspond, respectively, to pre- and post-bleaching time points. $t_{50}$ corresponds to the last recorded time point (t = 50 s). Images were color-coded using the Rainbow RGB function of ImageJ. Scale bar in **c**: 5 μm. **d** Average FRAP curves obtained after photobleaching of GFP-positive particles from brain explants. The following numbers of granules were analyzed: G080-GFP-Imp: 44; G080-GFP-Imp-CRISPR-ΔPLD: 46. **e** Average FRAP curves obtained after photobleaching of GFP-positive particles from S2R+ cells. The following numbers of particles were analyzed: GFP-Imp: 55; GFP-Imp-scr2: 57; GFP-Imp-scr4: 51. Error bars in **b**, **d**, **e** indicate s.e.m. **\*\***$P < 0.01$ (Mann–Whitney test on the distributions of normalized intensity values at t = 45 s (**b**) or 50 s (**d**)). Source data are provided as a Source Data file

adaptability and reversibility to the metastable protein interaction networks characteristic of such assemblies[30,59]. To date, however, the precise molecular requirements underlying PLD functions are still largely unclear. Work performed on the PLD of the yeast Nab3 protein has uncovered that some heterologous PLDs, but not all, can compensate for the lack of Nab3 PLD, suggesting the existence of distinct functional classes of PLDs[60]. No correlation could, however, be established in this study between PLD amino acid composition and/or length and functionality. More recent work performed on the PLDs of FUS-related RNA-binding proteins has decoded the molecular grammar underlying the phase

transitioning of these proteins, showing that interactions between tyrosine residues from PLDs and arginine residues from RNA-binding domains drive phase separation[61]. Such a grammar is not applicable to Imp PLD as this domain does not contain any tyrosine, leaving open the question of the molecular signature required for its function. Aligning Imp PLD sequences from different *Drosophila* species reveals some variations in primary sequences, in particular an increased length of glutamine repeats in distantly-related species, such as *D. grimshawi* or *D. mojavensis* (Supplementary Fig. 9a). To test if providing disorder was the main function of Imp PLD, or rather if PLD sequence was

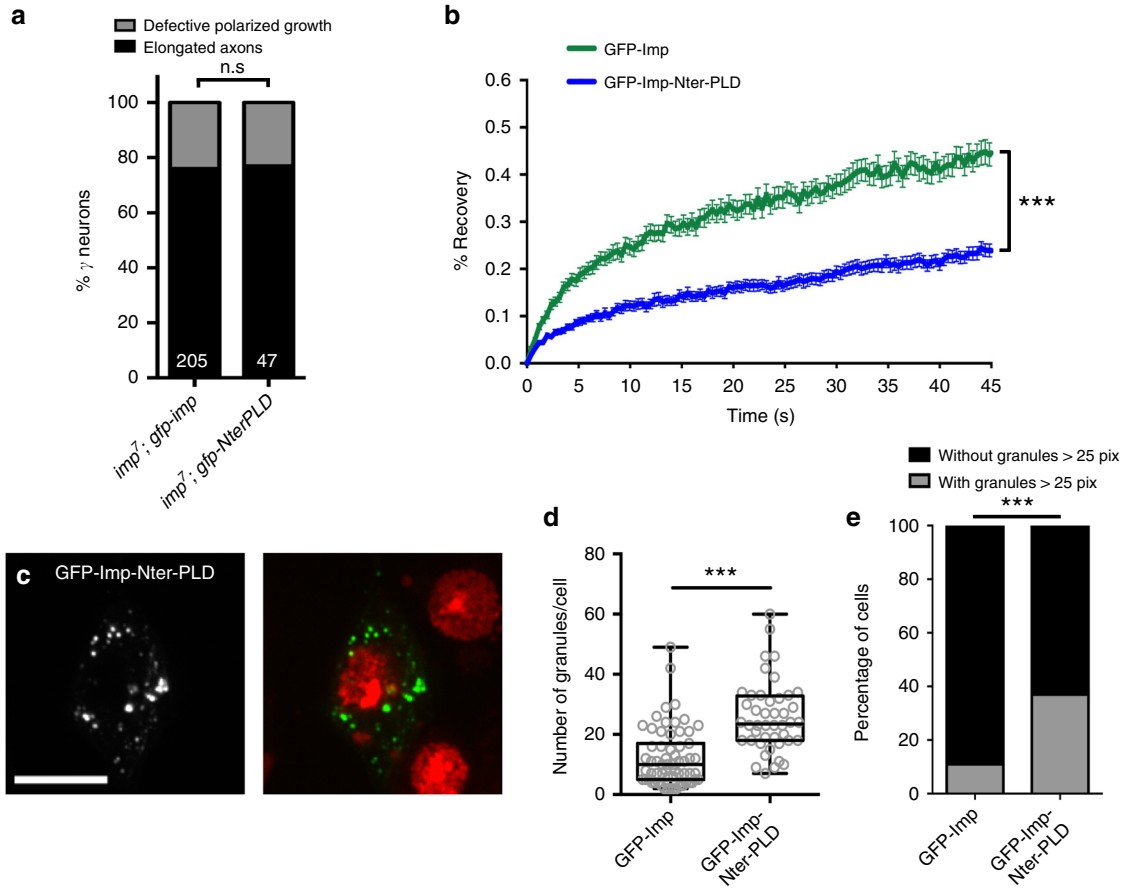

**Fig. 9** Imp PLD promotes axon regrowth independently of its role in granule properties. **a** Percentages of adult γ axons that succeeded (elongated axon) or failed (defective axonal growth) to reach the extremity of the medial lobe in MARCM experiments. n.s stands for not significant (Fisher's exact test). Numbers correspond to the total numbers of scored individual axons. Complete genotypes: FRT19A, tub-Gal80, hsp-flp/FRT19A $imp^7$; 201Y-Gal4, UAS-GFP/UAS-*gfp-imp* or UAS-*gfp-NterPLD*. **b** Average FRAP curves obtained after photobleaching of GFP-positive particles from S2R+ cells. The following numbers of particles were analyzed: GFP-Imp: 55; GFP-Imp-Nter-PLD: 56. Error bars indicate s.e.m. ***$P < 0.001$ (Mann–Whitney test on the distributions of normalized intensity values at t = 45 s). **c** S2R+ cells transfected with the GFP-Imp-Nter-PLD construct, and stained with DAPI (red in the overlay). GFP signal is shown in white (left) and green in the overlay. See Fig. 7b or Supplementary Fig. 7a for a control. Scale bar in **c**: 10 μm. **d** Distribution of cells in function of their number of GFP-Imp (left) or GFP-Imp-Nter-PLD (right) granules. ***$P < 0.001$ (Mann–Whitney test). The box plot is represented using the min to max convention, where the middle line defines the median and the whiskers go down to the smallest value and up to the largest. **e** Percentage of cells exhibiting granules larger than 25 pixels. ***$P < 0.001$ (Fisher's exact test). Seventy two and 71 cells were analyzed for GFP-Imp and GFP-Imp-Nter-PLD constructs respectively. Source data are provided as a Source Data file

encoding some information, we generated scrambled versions of Imp PLD that preserved overall amino acid composition but not primary sequence. Interestingly, scrambled PLDs could restrict granule assembly and regulate interaction dynamics as efficiently as wild-type domains, suggesting that primary sequence is not a determinant of Imp PLD function in granule regulation. How such disordered domains are regulated still has to be clarified, but both in vitro and in vivo work has shown that post-translational modifications of low-complexity domains (LCD), in particular phosphorylation of LCDs residues, can dramatically change phase separation behavior[24,62]. Imp PLD contain numerous serines, yet mutating all serines into non-phosphorylable glycines did not impact on granule assembly and dynamics in S2R+ cells, nor on MB γ axon remodeling in developing brains (Supplementary Fig. 9b–d). Whether other post-translational modifications contribute to the regulation of Imp PLD, and to changes in granule properties, remains to be determined.

Our work has revealed that Imp PLD is both necessary and sufficient to promote Imp axonal localization in vivo, and uncovered a role for Imp PLD in the microtubule-dependent transport of axonal RNP granules. While previous work has shown that ALS-disease-causing point mutations in the low-complexity domain of the TDP-43 RNA-binding protein alter the transport of TDP-43 granules along the axons of cultured neurons[16,63], whether such changes resulted from observed alterations of granule dynamics and physical properties remained unclear. In this study, we have shown that Imp-Nter-PLD constructs, while altering granule properties, localize normally to axons, suggesting that the function of Imp PLD in axonal transport is uncoupled from that in the modulation of granule dynamics. As further revealed by our quantitative real-time in vivo imaging, Imp PLD appears to promote the motility of axonal granules, and to modulate the velocity of retrograde granules. Although we have not been able to detect physical interactions between Imp and microtubule-dependent motor components such as Kinesin-heavy chain or Dynein, these results suggest that Imp PLD may be important for efficient activation or recruitment of such molecular motors. Of note, our analysis of Imp axonal transport was based on the tracking of granules detectable by confocal microscopy and we cannot formally exclude that part of the axonally-localized Imp travels independently of these structures. Consistent with the importance of

granule assembly in axonal transport, however, we have shown that Imp proteins localize essentially as puncta in adult γ lobes (Fig. 3d, i), and that a form of Imp that cannot assemble into visible granules (Imp-KH1–4DD) is not efficiently recruited to axons (Fig. 4e and Supplementary Fig. 4d).

Remarkably, localization of Imp neuronal granules to axons is under the control of a precise developmental program, as Imp granules, restricted to cell bodies during larval stages, start translocating to axons specifically during metamorphosis[37]. An interesting possibility is that the flexible nature of Imp PLD mediates a switch triggering maturation into transport-competent granules in response to developmental signals. Our grafting experiment has shown that such a function is transferable to the hIMP1 protein that normally lacks a PLD. How then are vertebrate proteins normally transported to axons independently of a PLD is still unclear, but it is conceivable that their transport relies on two extra N-terminal RRM domains. These domains, indeed, are dispensable for RNA binding, but important for subcellular targeting in non-neuronal cells and for binding to the KIF11 motor[64,65]. This example thus illustrates the complementarity of mechanisms that may be used for transport of RNP complexes, and is particularly interesting in a context where the molecular principles underlying the physiological regulation of neuronal RNP granules are still largely unclear[8].

## Methods

**Subcloning of Imp coding sequences**. Imp-ΔPLD-coding sequence was amplified by PCR from EST SD07045, using the impCS_sense/KH34-Gtwy-RP couple of primers. To generate the pENTR_ImpKH1–4DD variant, site directed mutagenesis was performed on previously described pENTR_Imp[37] using primers listed in Supplementary Table 1. Scrambled v2 and v4 PLDs were produced by gene synthesis, PCR-amplified, and added in frame to the ΔPLD sequence by SOE PCR using the primers listed in Table S1. PLD-Nter was generated by SOE PCR using the primers listed in Supplementary Table 1. hIMP1 was amplified from cDNA (gift from J. Chao) using the hIMP1-Gtwy-FP/RP primers, and hIMP1-drosPLD was generated by SOE PCR. The exact sequence of all above-mentioned primers is detailed in Supplementary Table 1. All sequence variants were subcloned into pENTR-D/TOPO vector (Life Technologies), fully sequenced, and recombined into Gateway destination vectors to express N-terminally-tagged proteins. pAGW (Murphy lab) and pUASt-attB-GFP[66] destination vectors were used for expression in S2R+ cells and Drosophila respectively.

**Expression of GFP-Imp variants in S2R+ cells and particle quantification**. S2R + cells were plated in six-well plates at a density of $5 \times 10^6$ cells per well, and incubated for 1 day at 25 °C. Cells were then transfected with 600 ng plasmids using Effectene (Qiagen). After 12 h, cells were resuspended in 1 mL of Schneider's medium supplemented with 10% Fetal Bovine Serum and Penicillin/streptomycin (1%), and transferred 24 h later to chambered Lab-Tek slides (four chambers; 250 μL per chamber). Cells were then fixed in 4% paraformaldehyde for 10 min, washed and permeabilized in PBS/0.1% Triton (PBT), and stained with DAPI. For detection of endogenous proteins, S2R+ cells were blocked in PBT supplemented with 1% BSA and then incubated overnight with rabbit anti-Imp antibodies (1:500[37];) in PBT supplemented with 0.1% BSA. Cells were then washed with PBT, incubated with Alexa488 conjugated anti-rabbit antibody (1:500; ThermoFisher A-21206) in PBT/0.1% BSA for 1–2 h, and washed once before DAPI labeling (5 min, 5 μg.mL$^{-1}$). After two washes in PBT, cells were mounted in Vectashield for imaging. Images were acquired on a Spinning Disc confocal microscope equipped with a Yokogawa CSU-X1 confocal head, and a iXON DU-897-BV EMCCD camera (Andor technology), using a UPLSAPO 100X oil 1.4 NA objective. Cells with low expression levels were selected for imaging and images were acquired using identical settings. As measured by western blot performed on S2R+ cell extracts, total amounts of GFP-Imp proteins were similar to those of endogenous Imp.

Granule size and number analysis were done on Maximum Intensity projections, and analysis performed using the SPADE algorithm (https://raweb.inria.fr/rapportsactivite/RA2016/morpheme/uid13.html), with manual edition for particles larger than 25 pixels. Minimal granule size was set to four pixels.

**Immunostainings and imaging of adult brains**. Adult fly brains were dissected in cold PBS, fixed for 25 min in 4% formaldehyde in PBS-0.1% Triton (PBT), washed three times in PBT and blocked overnight at 4 °C in PBS-0.3%Triton supplemented with BSA (1%). Brains were then incubated overnight at 4 °C with primary antibodies. The following antibodies were used: rat anti-Imp (1:1,000[37];), rabbit anti-GFP (1:1,000; Molecular Probes, A-11122), mouse anti-FasciclinII (1:15; DSHB,

1D4 clone), rabbit anti-Rin (1:200, gift from E. Gavis), rabbit anti-eIF4G (1:1000, gift from E. Izaurralde), rabbit anti-Rpl32 (1:1000; gift from M. Henze); mouse anti-FMRP (DSHB, 2F5–1 clone, 1:50), rabbit anti-eIF4e (gift from E. Izaurralde), rat anti-Pur-α (1:50; gift from K. Förstemann), rat anti-Staufen (1:1000, gift from A. Ephrussi), rabbit anti-Me31B (1:500, gift from C. Lim), rabbit anti-Tral (1:1000, gift from A. Nakamura), rabbit anti-Gawky (1:1000; gift from E. Izaurralde). Brains were then washed three times in PBT and incubated with fluorescently-coupled secondary antibodies. After four washes, brains were mounted in Vectashield (Vector labs, CliniSciences) and imaged with a LSM710, using a APO 40x NA 1.1 water objective for axon imaging, and a Plan Apo 63X NA 1.4 oil objective for cell body imaging. To assess the localization of UAS-GFP-Imp variants in γ axons, GFP intensity was measured from a 41 μm$^2$ region of interest located in the distal part of the γ lobe (selected using FasciclinII staining as a template), subtracted from the intensity of a neighbouring background region, and normalized.

For detection of endogenous GFP fluorescence, 1–3 day old female flies (G080-GFP-Imp; 201Y-Gal4,UAS-mCD8-RFP and G080-GFP-Imp-CRISPR-ΔPLD; 201Y-Gal4,UAS-mCD8-RFP) were dissected in Schneider's medium, fixed with 4% formaldehyde, and washed three times with 0.1% PBs-Triton (PBT). Images were acquired on freshly mounted samples, with a Zeiss LSM780 NLO inverted confocal microscope equipped with a GaAsP spectral detector and a Plan Apo 40 × 1.2 NA water objective for axons, and a 63 × 1.4 NA oil objective for cell bodies. Measurement of endogenous GFP-Imp axonal signal was performed as explained above (using the 201Y > CD8-RFP marker as a template). To assess the fluorescence intensity of Imp in the cell body of MB γ neurons, GFP-Imp endogenous fluorescence was measured in a 332 μm$^2$ region of a single confocal section and background intensity subtracted. Average intensities were normalized to 1 for controls.

**FRAP experiments and analysis**. FRAP experiments were performed on a Zeiss LSM780 NLO inverted confocal microscope, using the GaAsP spectral detector. Samples were imaged for five consecutive time frames and then bleached with 5 scan iterations (100% of a 35 mV 488 nm laser line). Fluorescence recovery was measured every 400 ms (for cells) and 1 s (for brains) for 115 time points.

For FRAP on transfected S2R+cells, cells were resuspended in 1 mL complete Schneider's medium after 24 h of expression, and plated in Nunc Lab-Tek II chambered Coverglass (two wells). Imaging was performed using a Plan Apo 63X oil 1.4 NA objective, and a circular region of interest of nine-pixel diameter was bleached (pixel size: 0.15 μm). A maximum of two granules were recorded per cell. For FRAP on brains, 12–14-day-old female flies were dissected (G080-GFP-Imp; 201Y-Gal4,UAS-mCD8-RFP and G080-GFP-Imp-CRISPR-ΔPLD; 201Y-Gal4, UAS-mCD8-RFP), their brains were mounted in a Labtek II chambered coverglass (#155378, Fisher Scientific) in culture medium (Schneider medium, 10% FCS, 1% Antibiotic Antimycotic Solution (Sigma), 200 μg.mL$^{-1}$ insulin (Sigma))[56], and left at room temperature for 1 h before imaging. Imaging was performed using a Plan Apo 40X water 1.2 NA objective, and a circular region of interest of 6 pixel diameter was bleached (pixel size: 0.083 μm). A maximum of four granules was recorded per hemisphere.

To quantify granule intensity over time, we manually tracked granules using the Fiji Manual Tracking plugin and used the xy coordinates of tracked particles to define ROI centers for each time points. Fluorescence intensities of the defined ROIs (three-pixel diameter) were measured using the Fiji Measure Track plugin (Chris Nicolai; http://rsb.info.nih.gov/ij/plugins/measure-track/index.html). Average intensities were normalized to pre-bleach intensities and corrected for acquisition bleaching (double normalization).

**Particle imaging and tracking**. For particle imaging on brains, 24 h APF pupae were dissected (G080-GFP-Imp; 201Y-Gal4, UAS-mCD8-RFP and G080-GFP-Imp-CRISPR-ΔPLD; 201Y-Gal4,UAS-mCD8-RFP). Their brains were mounted in a Labtek II chambered coverglass (#155378, Fisher Scientific) in culture medium (Schneider medium, 10% FCS, 1% Antibiotic Antimycotic Solution (Sigma), 200 μg.mL$^{-1}$ insulin (Sigma), 1 μg/ml ecdysone (20HE; Sigma))[56], and left at room temperature for 1 h before imaging. Experiment was performed on a Zeiss LSM880 Fast Airy Scan inverted confocal microscope, using the Fast Airy scan super resolution mode and a 40X water NA 1.1 objective. Stacks of three images (z step: 0.5 μm) were acquired every 1.2 s and image analysis done on maximum intensity projections. The images were corrected for bleaching using the Fiji Bleach correction > Histogram Matching method plugin.

All moving particles detected in MB peduncles were manually tracked using the Fiji Manual Tracking plugin, and each track was then split into runs, pauses, and reversals. Pauses were defined as events when the absolute value of instantaneous velocity dropped <50 nm.s$^{-1}$ for at least two consecutive frames[67]. Bidirectional granules were defined as granules undergoing at least one reversal that lasts for at least three consecutive steps. For analysis of run properties, runs were extracted from anterograde and retrograde granules, as well as from bidirectional granules for which both anterograde and retrograde components could be clearly assigned. Mean velocities were calculated using the following formula: $\mathrm{Sqrt}((x_f - x_i)^2 - (y_f - y_i)^2)/(t_f - t_i)$, where $(x, y)_f$ and $(x, y)_i$ refers to, respectively, the positions of individual tracked granules at final ($t_f$) and initial ($t_i$) time points.

**Fly genetics and generation of fly lines.** Flies were raised on standard food at 25 °C. All UAS-GFP-Imp constructs were inserted into the attP40 landing site via PhiC31-mediated integration and crossed with the 201Y-Gal4 line (gift of K. Ito) for analysis of subcellular distribution. To generate the GFP-Imp-CRISPR-ΔPLD line, gRNAs were cloned into the pDCC6 plasmid using the CTTCGCAACA GCAACAGAGCCTAGC and AAACGCTAGGCTCTGTTGCTGTTGC sense and antisense primers, and injected into G080/+; attP40-nos-Cas9 embryos together with a donor construct containing a 3XP3-RFP selection cassette flanked by LoxP sites and 5′ and 3′ homology arms. The 5′ homology arm was amplified using the ATTGAGAACATGTCGCGTGC and ttactaCTGTTGCTGTTGTTGCAATTGTT primers, and the 3′ homology arm using the AACAGCCACAGTCGCCATCT and ACGCTTTGCTCACTTCTCTTCT primers. RFP+ individuals were screened by PCR, and the G080-GFP-Imp-CRISPR-ΔPLD line validated by sequencing. MCFO experiments were performed using the HA_V5_FLAG cassette[49] and a hsp-flp inserted on the second chromosome; flies were raised at 18 °C. MARCM experiments were performed as previously described[51].

**Western blots.** For western blots, protein extracts were subjected to electrophoresis, blotted to PVDF membranes, and probed with the following primary antibodies: rabbit anti-GFP (1:2,500; #TP-401; Torey Pines); mouse anti-Tubulin (1:5,000; DM1A clone; Sigma). Original western blot images are shown in the Source data file.

**Circular dichroïsm.** The PLD of Imp was expressed in *E. coli* Rosetta2 bacterial cells transformed with pETM40-PLD containing a cleavable MBP N-ter tag and a His C-ter tag. The cells were sonicated in sonication buffer (20 mM Tris Hcl pH 7.6, 200 mM NaCl, protease inhibitors (Roche, 11836170001) and 1 mg.mL$^{-1}$ lysozyme (Sigma L6876)) and the lysate treated with turbo DNase (Ambion™). Soluble MBP-tagged proteins were captured using amylose resin (NEB), and eluted with elution buffer A (20 mM Tris HCL pH 7.6, 15 mM Maltose (Sigma M9171)). Recombinant proteins were then incubated with TEV, applied to a HI-TRAP TALON crude column (GE healthcare 10431065), eluted with elution buffer B (20 mM Tris HCL pH 7.6, 150 mM Imidazole (Sigma I2399)), and dialyzed into dialysis buffer (20 mM Tris Hcl pH 7.6, 5 0 mM NaCl and 20% glycerol).

Far ultraviolet circular dichroïsm spectra were recorded at 20 °C using a JASCO J810 dichrograph equipped with a thermostatted cell holder and 0.1 mm pathlength quartz cuvettes (Hellma, Müllheim, Germany). Each spectrum was the average of five acquisitions recorded at a speed of 50 nm min$^{-1}$, in 1 nm increments from 260 to 190 nm, and a bandwidth of 1 nm. All spectra were buffer-corrected and normalized to the mean residue weight ellipticity (θMRW; degrees × cm$^2$.dmole$^{-1}$) using the equation θ(λ)MRW = θ(λ)mdeg/10$cnd$, where θ(λ)mdeg is the recorded spectra in millidegrees, $c$ is the sample concentration in moles per liter, $n$ is the number of amino acid residues, and $d$ is the path length of the cuvette in centimeters.

**EMSA.** EMSA experiments were performed with recombinant GFP Imp and GFP Imp KH1–4DD proteins expressed using the baculovirus expression system. Briefly, Imp sequences were cloned into pocc29 plasmids containing a Nter GFP tag and a cleavable MBP C-ter tag. After expression, cells were lysed in resuspension buffer (50 mM Tris-HCl pH 8, 1 M KCl, 5% glycerol, 0.1% CHAPS, 1 mM DTT), recombinant proteins were captured using amylose resin (NEB), and eluted with elution buffer (resuspension buffer complemented with 10 mM maltose).

**Reporting summary.** Further information on research design is available in the Nature Research Reporting Summary linked to this article.

## Data availability

All relevant data are available from the corresponding author upon reasonable request. The source data underlying Figs. 1c, 2h, 3c, f, i, 4f, g, 6c, d, h, 7d, e, h, i, 8b, d, e, 9a, b, d, e and Supplementary Figs. 2a, 4a–d, 6a, c, e, f, 7d,f,g and 8a are provided as a Source Data file.

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

## Acknowledgements

This study was supported by ANR (ANR-15-CE12–0016) and Fondation ARC (PJA 20141201623) grants to F.B., and by the LABEX SIGNALIFE program (#ANR – 11 – LABX – 0028 – 01). J.V. was funded by fellowships from the French Ministry of Research and the ARC, as well as by Ecole Doctorale ED85 and Cancéropôle PACA travel grants. We thank L. Palin for excellent technical assistance, F. De Graeve and C. Medioni for their help with granule analysis and real-time imaging, K.V. Pushpalatha for her help with the analysis of GFP-Imp-ΔPLD-gfp start cells, R. Melki for hosting J.V. in his lab to perform circular dichroism, members of the Alberti lab and the Protein Facility at the MPI-CBG, Dresden for their assistance with protein expression and purification, and for helping to set up the HeLa cell experiments. We thank the PRISM Imaging Facility for use of their microscopes and support, especially M. Mertz, M. Mondin, S. Lachambre and S. Schaub for their help with live-imaging experiments. We are grateful to members of the Besse group and I. Gaspar for discussion and advise. We thank I. Gaspar, A. Hübstenberg and C. Medioni for critical reading of the manuscript. We are grateful to the Bloomington Drosophila Stock Center and the Developmental Studies Hybridoma Bank for reagents.

## Author contributions

J.V. performed all experiments and quantifications presented in the manuscript. C.P. generated the Imp-ΔPLD and hIMP1 variants and performed the initial analyses of Imp-ΔPLD proteins in S2R+ cells and fly brains. M.H. cloned the KH1–4DD, Nter-PLD, and Scr Imp variants and established the corresponding fly stocks. J.V. and L.B. performed the Circular Dichroïsm experiments. J.V., C.P., S.A., and F.B. contributed to hypothesis development, experimental design, and data interpretation. F.B. provided the overall supervision, the funding, and wrote the article. All authors discussed the data and commented on the manuscript.
