## [Peer Review File · Nature Communications]

Reviewers' comments:

Reviewer #1 (Remarks to the Author):

This work confirms the existence and describes the functions of the PLD of Imp, a RNP granule protein in drosophila. Unlike other PLDs recently studied in the field, such as those in RNA-binding proteins hnRNPA1, FUS, and TDP-43, Imp PLD is not important for its recruitment to liquid-like RNP granules. It is the folded KH domains, through their interactions with RNA, that appear to drive the localization of Imp to RNP granules. Instead, the presence of PLD in Imp reduces the number and size of RNP granules but enhances granule dynamics. Furthermore, the position (but not the primary structure) of this PLD is important for its role in modulating RNP homeostasis, such as granule size and number. The authors also found a second, position-independent role for PLD in modulating Imp granule axon localization, motility, and transport that builds on prior work that showed that Imp was a key regulator of axonal remodeling. This study suggests new regulatory functionality for PLDs that could exist in other proteins, particularly those that undergo liquid-liquid phase separation (LLPS). It will be interesting to determine the molecular basis by which the PLD regulate Imp granule function in future work. This thorough, well-executed study presents new and exciting findings that are worthy of publication in Nature Communications after the following questions are addressed.

Questions:

The average size of GFP-Imp-deltaPLD granules is only slightly larger than GFP-Imp granules (9 vs 8 pixels), despite large granules (>25 pixels) being present. Can the authors reconcile this observation? Furthermore, did the authors FRAP these different-sized granules (small vs. large) to determine if a relationship exists between granule dynamics (i.e. FRAP recovery rates and mobile/immobile fractions) and granule size?

Could the authors comment on granule size of GFP-hIMP1 and GFP-hIMP1-drosPLD as noted for drosophila Imp granules (see top of pg 7)?

What is the impact of endogenous Imp protein on the transfections of Imp constructs? Can the authors report on the relative amount of transfected Imp vs. endogenous Imp present by Western Blot?

The authors point out that Imp does not colocalize with stress granule markers. How were stress granules induced in S2R+ cells?

The observation that PLD has a negative effect on granule size and number in cells lead to some interesting experiments that could be performed in a reconstituted in vitro system, but may be beyond the scope of this work. It would be interesting to determine whether full-length Imp protein forms droplets or undergoes LLPS in vitro with RNA. Similarly, if the deltaPLD construct leads to increased droplet size and number, that would provide for a rich in vitro system to test the molecular function of the PLD. It would be interesting to determine if the presence of the PLD reduces the propensity for Imp to phase separate with RNA, or that the presence of the PLD alters droplet morphology or dynamics, or a combination of both.

Are there any implications for why human Imp does not contain a PLD, but drosophila does? Are there other Imp homologs (not drosophila) with similar PLDs?

In Figure 6, it would be useful to outline the medial lobe and the expected MB gamma neuron axonal bundle where signal should be expected for GFP-Imp. It may be useful to the reader if Figure S6A was moved to the main text, and if the gamma neurons were identified in this figure. Could the authors outline these neurons, or show an image similar to Figure 1C from Medioni et al. Current Biology 24, 793-800 (2014)? The authors further point out a 'strong reduction in the localization of GFP-Imp proteins to MB gamma axons ... in the absence of Imp PLD' in Figure 6D-F.

Can the authors more clearly define the axons in Figures 6D-E? The periphery of Figures 6D-E seem to contain cell bodies, is this correct?

In Figure 8, I am confused by the data for wt and the rescues made by the Imp constructs. For wt, I would have assumed that axons all elongate and reach the extremity of the medial lobe? In other words, is the legend switched such that the black bar represents 'elongated axons'? Based on the results section on pg 11, I would expect that gfp-Imp would restore elongation functionality rather than increase defective axonal growth.

Minor issues:

Pg 11 Reference to fig 8J-L should be to Figure 6.

Please show the full WB for Figure S3B, S3C.

Information regarding Imp antibody source is missing from the Methods section on pg 16.

Reviewer #2 (Remarks to the Author):

This study from Florence Besse's lab studies properties of a prion-like domain (PLD) in the *Drosophila* Imp protein, an RNA-binding protein which the lab has previously studied in the context of axonal remodeling in the *Drosophila* mushroom body (<https://doi.org/10.1016/j.cub.2014.02.038>). In this manuscript the authors study the contribution of the PLD to the properties (dynamics, size and transport) of granules containing Imp in both cultured cells and in the *Drosophila* brain. The paper has some potential: there is currently a lot of attention in the field on biochemical/biophysical properties of PLDs and their role in the formation of membrane-less organelles – granules. However it has been challenging to study the biological function of granules, particularly in an in vivo context. I was therefore at the outset intrigued to read the paper with an expectation from the abstract that it would link the granules and/or PLD domain to functions in localization of RNA to axons and axon outgrowth, which I believe at this point are novel biological functions for granules. The paper takes steps in this direction, however as currently framed it is more focused and bogged-down with abstract properties of the granules – their dynamics, size, number, and properties of their motility. For most of these properties there is no ascribed biological relevance, nor is there mechanism, since information on the composition of these objects in cells is lacking. I gained some hope at the end of the paper (Figure 8), where the authors finally do present a functional assay for Imp in axon outgrowth in the *Drosophila* mushroom body. The phenotype of imp mutants alone is modest but significant when assessed in a mosaic genetic background, and the authors are able to partially rescue this phenotype with GFP-Imp but not with GFP-Imp deleted for the PLD domain. This implies that the PLD domain has a biological function that can be studied further, and opens the possibility that this functional assay may be probed for correlation with the other studied properties (dynamics and transport) of the granules. Based on the data shown thus far in the paper, the PLD domain allows for axonal localization of Imp, and this correlates with its function in axonal outgrowth. This may alone be a smaller and more clear story. Whether or not the altered granule characteristics, which are currently a major focus of this paper, have any biological function is not clear, and thus far there is actually some negative data for this with the N-terminal PLD construct, which rescues axon outgrowth similarly to the wild type GFP-Imp protein.

Some specific comments:

Even the title of the paper focuses on 'functions' of Imp in axonal transport and 'RNP granule homeostasis', whose biological meaning is not clear. I suggest that the authors put more emphasis on their ability to study biological function of Imp in axonal outgrowth, and then probe the relevance of transport and granule properties to its biological function(s). Rather than ending the paper with the rescue assay, I wonder why the authors don't start with it, so that the study can be

grounded in its biological context and premise that the PLD is an important domain for Imp's function in axonal outgrowth. There would still be significant work to follow through with this framework, and the results may turn out to be of medium (not high) impact, but the project may obtain more interest and significance if grounded in this way.

The biochemical makeup of the granules is not characterized, and may also be changing in the Imp mutations that are probed, but this is not assessed. If the authors want to describe altered properties of the granules in cells and in neurons they need to have more information about these granules. For instance, the data presented may be explained by the presence of additional PLD containing proteins in the granules. Also, from their previous study the authors propose that the granules contain profilin mRNA. However the profilin RNA localization in Figure S2C and S2D is actually strikingly in bright puncta that do NOT co-localize with profilin mRNA. While profilin mRNA can be detected by sensitive QRT-PCR as co-immunoprecipitating with Imp, this is only a minor fraction of the profilin mRNA in cells. Conversely profilin mRNA may be only a minor fraction of potential RNAs or other proteins associated with Imp. With these issues unaddressed, the granule properties described in the paper are lacking in both mechanism and potential meaning.

The data presented in Figure 1 are strikingly thin. It is not usual to see one circular dichroism plot with no controls or conditions for comparison. Moreover, since the authors have purified Imp protein in hand it is puzzling that they do not examine its ability to form granules and phase separations in vitro. In this case they can actually know the composition, for comparison/contrast the granules imaged in cells, whose composition is not known.

Other more minor points:

The data in Figure 6, which are meant to show localization of GFP-Imp to MB axons is not well oriented: its very hard to see from the images what is interpreted to be 'axonal' localization and what is quantified. The text refers to some additional orientation panels in Figure S6, which is not present in this manuscript. Further orientation needs to be presented in the main figures.

Figure S6 was not present in the submitted material. Also there is a reference in error (on p.11) to Figure panels 8J-L, which are also not present, however I think the authors mean to refer to Figure 6J-L.

Reviewer #3 (Remarks to the Author):

The manuscript explores the role of a prion-like domain (PLD) of the RNA binding protein IMP in the formation of IMP positive granules (in cells and in vivo) and in axon migration in the *Drosophila* brain.

They show that the PLD is not necessary for granule formation. The fact that PLD does not drive granule formation and is not even necessary is in line with reports that have shown that prion like domains sometimes drive granule formation and sometimes only modulate it. It suggests that either PLD or IMP in its native conformation acts as a repellent for other molecules and has a higher dynamics.

However, the PLD also appears to have a positive role because whether at the Cter (as in WT IMP) or Nter, it rescues the lack of IMP in terms of mushroom body recruitment, granule mobility and axon guidance. In contrast, IMP without PLD does not.

The ms represents a lot of interesting work of high quality. However, the positive role of the PLD is understudied at the molecular level. Please see my specific points.

1) The manuscript first explores the role of a prion-like domain (PLD) of the RNA binding protein IMP in the formation of IMP positive granules in *Drosophila* cells (S2 cells) and in vivo in the

Drosophila mushroom body.

The authors show that this PLD is not necessary for granule formation in either system, but that PLD modulates down the number and size of the formed particles. Without it, IMP is more efficiently recruited to granules in cells and mushroom body. Conversely, when PLD is appended to human IMP1 (that does not display a PLD), granule formation is slightly impaired (at least in cells).

One line of explanation for this finding is that the presence of PLD makes the IMP molecules more mobile in and out of granules (FRAP). The recovery of full length Imp is more efficient and higher when compared to the form lacking PLD. This is further exemplified the higher background showing that the IMP molecules are overall less recruited (compare Figure 1 B versus D)

This figure should be further improved by performing more analysis.

1.1: The number of granules in mushroom body should be estimated.

In Figure 1G, GFP-IMP forms a very small number of granules, less than endogenous IMP. This should be quantified.

1.2: Is there also a dynamic effect in granule formation due to the presence of PLD. Can a time course be performed? DO granules form faster in cells expressing delta PLD IMP?

This could be monitored if the expression vectors were not inducible.

1.3: Is the frequency of cells displaying granules the same in all conditions? This should be mentioned on top of the number of granules and their size? Is there also more cells positive for granule formation?

2) Figure 4: The authors argue that the position of PLD is critical for this function as a repellent. Scrambled PLD-IMP has the same granule features as WT IMP (PLD at the C-ter). However, when PLD is at the N-ter, it acts as a form of IMP lacking PLD (larger and more granules and less background).

The reviewer argues this effect is not related to the position of the PLD. The reviewer argues that the salient feature is the absence of a sequence (any sequence) at its C-ter. Without it, IMP conformation is compromised and prevents its dynamics. Addition of PLD is at the N-ter is irrelevant because the C-ter is not occupied.

Can the authors use a method (perhaps CD) to monitor IMP conformation with or without PLD at the Cter. This is a critical experiment. It is possible that the folded WT IMP hides sticky pieces and display a different conformation. In this regard, the review took the liberty to model the structure of full length IMP, and IMP without PLD using available softwares and the conformation appears different quite different. This needs to be strengthened.

An alternative manner to evaluate a change in the IMP conformation when it lacksthe PLD at Cter is to assess whether the binding affinity to mRNA as in figure 1.

Furthermore, identifying the interactome of full length IMP and IMP lacking PLD reveal possible important differences

3) In this context, the experiment presented in Figure 4 should be completed by 3 additional tests.

3.1: IMP C-ter should be occupied by an irrelevant sequence, very different from PLD Afterall, scrambling PLD that is so rich in Q will end up in rather similar sequences. What about a totally different sequence of the same length?

Will this behave a WT IMP or delta PLD IMP?

3.2: Can this chimera (or the one with Scrambled PLD) be appended with real PLS at the N-ter?

If the conformation is rescued with addition of any sequence at the C-ter, perhaps a role for PLD in granule formation can be discovered in line with its role in axons independent of granule dynamics.

3.3: In this context, the human IMP1 should be appended by scrambled dros PLD, not the WT form. Again, addition of the C-ter might fold the protein differently and be enough to affect granule formation.

4) The authors monitor IMP recruitment to the mushroom body and show that without PLD, IMP is not present in the mushroom body.

The reviewer is not an expert but he wonders how can Figure 1 J be achieved if delta PLD-IMP cannot reach the mushroom bodies.

5) Interestingly, N-ter PLD is recruited to the MB, and so is hIMP1-dros PLD. This suggests a positive role for PLD on top of occupying the C-ter. Is this related to the conformation? (see above).

6) Figure 7: The granule dynamics is virtually similar (with the speed of delta PLD-IMP + positive granules slightly higher than full length IMP). What does it mean?

Would Nter PLD-IMP act as WT-IMP or delta PLD-IMP as suggested by experiments presented in Figure 8?

7) Sadly, the positive role of PLD in recruitment to MB and axon morphology is understudied and is preliminary.

Has any interactors of PLD be determined perhaps by mass spec that could explain this role?

If the delta PLD has a different conformation, the mass spec experiment will need to be carefully controlled to pick up specific interactors (see also point above).

Reviewer #1 (Remarks to the Author):

This work confirms the existence and describes the functions of the PLD of Imp, a RNP granule protein in drosophila. Unlike other PLDs recently studied in the field, such as those in RNA-binding proteins hnRNPA1, FUS, and TDP-43, Imp PLD is not important for its recruitment to liquid-like RNP granules. It is the folded KH domains, through their interactions with RNA, that appear to drive the localization of Imp to RNP granules. Instead, the presence of PLD in Imp reduces the number and size of RNP granules but enhances granule dynamics. Furthermore, the position (but not the primary structure) of this PLD is important for its role in modulating RNP homeostasis, such as granule size and number. The authors also found a second, position-independent role for PLD in modulating Imp granule axon localization, motility, and transport that builds on prior work that showed that Imp was a key regulator of axonal remodeling. This study suggests new regulatory functionality for PLDs that could exist in other proteins, particularly those that undergo liquid-liquid phase separation (LLPS). It will be interesting to determine the molecular basis by which the PLD regulate Imp granule function in future work. **This thorough, well-executed study presents new and exciting findings that are worthy of publication in Nature Communications after the following questions are addressed.**

Questions:

- The average size of GFP-Imp-deltaPLD granules is only slightly larger than GFP-Imp granules (9 vs 8 pixels), despite large granules (>25 pixels) being present. Can the authors reconcile this observation? We thank the reviewer for pointing that out. Removing the PLD of Imp increases the total number of granules, increasing the number of both small and large granules. Although removing Imp PLD also generates abnormally large granules, such granules represent only a small fraction of the total, and thus only slightly shift the average value. We realized that comparing average values does not accurately reflect the distribution of granule sizes and have thus removed these values from the text. We have replaced them by a frequency distribution plot (Figure I and Supplementary Figure 6B of the revised manuscript) that we now comment in the main text.

Figure I. Frequency distribution of the size of GFP-Imp and GFP-Imp-ΔPLD granules.

This graph is displayed in Supplementary Figure 6C of the revised manuscript.

Furthermore, did the authors FRAP these different-sized granules (small vs. large) to determine if a relationship exists between granule dynamics (i.e. FRAP recovery rates and mobile/immobile fractions) and granule size?

To first address whether granule dynamics may depend on granule size, we performed FRAP on “large granules” of a constant diameter (9 pixels^{*}) for both GFP-Imp and GFP-Imp-ΔPLD constructs (Figure II). In this context, we observed that fluorescence recovery is lower for GFP-Imp-ΔPLD granules than for GFP-Imp granules, indicating that the difference we initially reported does not result from increased average granule size. To further confirm this point, we compared the fluorescence recovery of small (approximately 5 pixels) granules formed by GFP-Imp and GFP-Imp-ΔPLD proteins. Although fluorescence recoveries were in general higher for small granules, a significant difference was still observed in the GFP-Imp-ΔPLD conditions, further suggesting that granule size is not directly influencing the dynamics of GFP-Imp-ΔPLD granules. Corresponding graphs are now included in Supplementary Figure S8A and results mentioned in the main text.

^{*} Note that pixel sizes are different for FRAP experiments and quantification of granule size

Figure II. FRAP recovery curves for small and large granules in S2R+ cells.

Note that a decreased fluorescence recovery is observed in the absence of Imp PLD for both small and large granules. This graph is displayed in Supplementary Figure S8A of the revised manuscript.

Could the authors comment on granule size of GFP-hIMP1 and GFP-hIMP1-drosPLD as noted for drosophila Imp granules (see top of pg 7)?

As explained above, we realized that average values do not reflect the diversity of granule sizes. In addition to the graph displayed in Figure 7I, we thus now include in Supplementary Figure S6F a frequency distribution plot of hIMP1 and hIMP1-drosPLD granule sizes (see also Figure III).

Figure III. Frequency distribution of the size of GFP-hIMP1 and GFP-hIMP1-drosPLD granules.

This graph is displayed in Supplementary Figure S6F of the revised manuscript.

What is the impact of endogenous Imp protein on the transfections of Imp constructs?

We have tried in S2R+ cells to downregulate the expression of endogenous *imp* using dsRNAs targeting the 3'UTR sequence of *imp* (which is absent from the transfected GFP-constructs). Although we tried using different dsRNA sequences, different amounts, and increasing incubation times, we have not been able to significantly downregulate the expression of endogenous Imp (see Figure IV for an example). This might be due to the fact that Imp is binding to its own transcripts, and thus possibly involved in an auto-regulatory loop.

Figure IV. Inefficient downregulation of Imp levels upon RNAi.

Western-Blot performed using protein extracts from S2R+ cells incubated for 5 days with *gfp* dsRNA (control) or two independent *imp* dsRNA (#1 and #2). Tubulin was used as a loading control.

Our *in vivo* data however unambiguously demonstrate that GFP-Imp- Δ PLD variants behave similarly in the presence (201Y-Gal4, UAS-GFP-Imp- Δ PLD) and in the absence (CRISPR-G080-Imp- Δ PLD flies) of endogenous *imp*.

Can the authors report on the relative amount of transfected Imp vs. endogenous Imp present by Western Blot?

We have performed Western-Blots on extracts of S2R+ cells transfected with the GFP-Imp construct. As shown in Figure V, total amounts of GFP-Imp proteins were in the same range as those of endogenous Imp. This information is now mentioned in the Materials and Methods section. Of note, and as specified in this section, we only imaged cells with similar low expression levels of GFP-

constructs for analysis.

Figure V. Relative amounts of GFP-Imp and endogenous Imp in S2R+ cell extracts.

Two different amounts were loaded on the gel. The membrane was blotted with anti-Imp (left; red in the overlay) and anti-GFP (green in the overlay) antibodies. Note that some variability was observed in the value of the GFP-Imp:Imp intensity ratio.

The authors point out that Imp does not colocalize with stress granule markers. How were stress granules induced in S2R+ cells?

We thank the reviewer for pointing out that our text was ambiguous. We have now clarified this point, and better explained i) that the Imp granules we study are detected in the absence of stress, and ii) that expressing GFP-Imp fusions does not induce the formation of stress granules (see Supplementary Figures 3A and B for analysis in brains, and Supplementary Figure 5A,B for analysis in cells).

The observation that PLD has a negative effect on granule size and number in cells lead to some interesting experiments that could be performed in a reconstituted *in vitro* system, but **may be beyond the scope of this work**. It would be interesting to determine whether full-length Imp protein forms droplets or undergoes LLPS *in vitro* with RNA. Similarly, if the deltaPLD construct leads to increased droplet size and number, that would provide for a rich *in vitro* system to test the molecular function of the PLD. It would be interesting to determine if the presence of the PLD reduces the propensity for Imp to phase separate with RNA, or that the presence of the PLD alters droplet morphology or dynamics, or a combination of both.

We have produced and purified recombinant GFP-tagged Imp proteins (both wild-type and Δ PLD proteins) from Sf9 insect cells, using high quality standards established in the Alberti lab, and have used these proteins in reconstituted *in vitro* assays. While Imp-wt and Imp- Δ PLD did not assemble into visible structures when diluted into physiological salt concentrations, both proteins assembled into small (<1 μ m) assemblies in the presence of a molecular crowder (either 10% dextran or 10% ficoll) (Figure VI, left and data not shown). Interestingly, Imp- Δ PLD proteins matured differently than Imp-wt proteins, assembling into elongated structures after 4h (Figure VI, middle and right). However, Imp-wt and Imp- Δ PLD assemblies did not fuse into bigger droplets with time, nor did they respond to increasing salt concentration (from 50 mM up to 700 mM), or presence of RNA (5 nM) (data not shown).

Figure VI. Behavior of Imp-wt and Imp- Δ PLD recombinant proteins in *in vitro* reconstituted systems.

Experiment performed with 0.5 μ M protein, 150 mM KCl and 10% Ficoll. Note that GFP-Imp- Δ PLD proteins had a tendency to mature into elongated structures. Boxed regions are shown at a higher magnification on the right.

Why these assemblies did not recapitulate the properties of RNP complexes is unclear, but it may be that Imp needs the presence of specific partners or to be post-translationally modified. This complicates interpretation of these *in vitro* results, and we would therefore prefer to not mention them. As highlighted by the referee, the main objective of our study was to highlight the cellular and physiological function of the PLD domain, and we agree with him/her that *in vitro* assays are outside the main scope of our work.

Are there any implications for why human Imp does not contain a PLD, but drosophila does? Are there other Imp homologs (not drosophila) with similar PLDs?

The reviewer raised an interesting point. As shown in different contexts, PLDs evolve fast and are not necessarily conserved (both in term of sequence and in term of presence/absence). Still, the lack of a PLD in the vertebrate proteins raises the question of how these proteins are transported to axons in vertebrate neurons. In our revised discussion, we propose that the lack of a PLD may be compensated by the presence of 2 extra RRM domains known to be dispensable for RNA binding, but involved in the interaction with the Kif11 motor^{1,2}. This nicely illustrates how complementary mechanisms may be used for transport of RNP complexes.

With regard to the presence of a similar PLD in other species, we know that nematode Imps contain a long low-complexity domain. Whether this domain has a conserved function is unclear, especially as its primary sequence is quite divergent from the *Drosophila* PLD. For this reason, we are not mentioning this point in the discussion.

In Figure 6, it would be useful to outline the medial lobe and the expected MB gamma neuron axonal bundle where signal should be expected for GFP-Imp. It may be useful to the reader if Figure S6A was moved to the main text, and if the gamma neurons were identified in this figure. Could the authors outline these neurons, or show an image similar to Figure 1C from Medioni et al. Current Biology 24, 793-800 (2014)? The authors further point out a ‘strong reduction in the localization of GFP-Imp proteins to MB gamma axons ... in the absence of Imp PLD’ in Figure 6D-F. Can the authors more clearly define the axons in Figures 6D-E? The periphery of Figures 6D-E seem to contain cell bodies, is this correct?

As suggested by the referee, we have now included a schematic representation of MBs in the main Figure 2 (Figure 2A). We have also highlighted with dotted white lines the limits of the γ axon population in revised Figure 3, and mention in the legend to Figure 3D,E that the GFP-Imp+ cell bodies observed in the image belong to different populations.

In Figure 8, I am confused by the data for wt and the rescues made by the Imp constructs. For wt, I would have assumed that axons all elongate and reach the extremity of the medial lobe? In other words, is the legend switched such that the black bar represents ‘elongated axons’? Based on the results section on pg 11, I would expect that gfp-Imp would restore elongation functionality rather than increase defective axonal growth.

We thank the reviewer for pointing this out. Legends were indeed inverted and this mistake has been corrected in the revised version (now Figure 2H).

Minor issues:

Pg 11 Reference to fig 8J-L should be to Figure 6.

Figure order has been extensively changed in response to Referee #2, but we have ensured that references to the mentioned data are now correct.

Please show the full WB for Figure S3B, S3C.

Lanes had been extracted from the original gel as two different amounts had been loaded for each replicate. The full gels are now displayed in revised Supplementary Figures 6A and E.

Information regarding Imp antibody source is missing from the Methods section on pg 16.

Reference to the Medioni et al., 2014 paper has now been added.

Reviewer #2 (Remarks to the Author):

This study from Florence Besse’s lab studies properties of a prion-like domain (PLD) in the *Drosophila* Imp protein, an RNA-binding protein which the lab has previously studied in the context

of axonal remodeling in the *Drosophila* mushroom body (<https://doi.org/10.1016/j.cub.2014.02.038>). In this manuscript the authors study the contribution of the PLD to the properties (dynamics, size and transport) of granules containing Imp in both cultured cells and in the *Drosophila* brain. **The paper has some potential: there is currently a lot of attention in the field on biochemical/biophysical properties of PLDs and their role in the formation of membrane-less organelles – granules. However it has been challenging to study the biological function of granules, particularly in an *in vivo* context.** I was therefore at the outset intrigued to read the paper with an expectation from the abstract that it would link the granules and/or PLD domain to functions in localization of RNA to axons and axon outgrowth, which I believe at this point are novel biological functions for granules. The paper takes steps in this direction, however as currently framed it is more focused and bogged-down with abstract properties of the granules – their dynamics, size, number, and properties of their motility. For most of these properties there is no ascribed biological relevance, nor is there mechanism, since information on the composition of these objects in cells is lacking. I gained some hope at the end of the paper (Figure 8), where the authors finally do present a functional assay for Imp in axon outgrowth in the *Drosophila* mushroom body. The phenotype of imp mutants alone is modest but significant when assessed in a mosaic genetic background, and the authors are able to partially rescue this phenotype with GFP-Imp but not with GFP-Imp deleted for the PLD domain. This implies that the PLD domain has a biological function that can be studied further, and opens the possibility that this functional assay may be probed for correlation with the other studied properties (dynamics and transport) of the granules. Based on the data shown thus far in the paper, the PLD domain allows for axonal localization of Imp, and this correlates with its function in axonal outgrowth. This may alone be a smaller and more clear story. Whether or not the altered granule characteristics, which are currently a major focus of this paper, have any biological function is not clear, and thus far there is actually some negative data for this with the N-terminal PLD construct, which rescues axon outgrowth similarly to the wild type GFP-Imp protein.

Some specific comments:

1- Even the title of the paper focuses on ‘functions’ of Imp in axonal transport and ‘RNP granule homeostasis’, whose biological meaning is not clear. I suggest that the authors put more emphasis on their ability to study biological function of Imp in axonal outgrowth, and then probe the relevance of transport and granule properties to its biological function(s). Rather than ending the paper with the rescue assay, I wonder why the authors don’t start with it, so that the study can be grounded in its biological context and premise that the PLD is an important domain for Imp’s function in axonal outgrowth. There would still be significant work to follow through with this framework, and the results may turn out to be of medium (not high) impact, but the project may obtain more interest and significance if grounded in this way.

We thank the reviewer for his/her suggestion. As proposed, we have extensively remodeled the text and figures, have changed the title, and are now presenting the *in vivo* data as the starting point of the manuscript. Furthermore, we have extended our description of the *in vivo* phenotypes. First, we now provide data showing the collective (Figure 2B,C) and individual (Figure 2D,E) behavior of Imp- Δ PLD neurons. Second, we now analyze the molecular determinants underlying Imp PLD function in the *in vivo* context (Figures 3 and 4). Finally, we have reduced the part dedicated to the quantification of granule size and number in S2R+ cells (only one main Figure: Figure 7). We think that the referee’s suggestion indeed significantly improved the manuscript, and helped us highlight the main findings of this manuscript, that relate i) to the *in vivo* functions of Imp PLD and ii) to the fact that these functions are uncoupled from changes in granule characteristics.

2-The biochemical makeup of the granules is not characterized, and may also be changing in the Imp mutations that are probed, but this is not assessed. If the authors want to describe altered properties of the granules in cells and in neurons **they need to have more information about these granules**. For instance, the data presented may be explained by the presence of additional PLD containing proteins in the granules.

To better characterize the content of Imp RNP granules, and compare granule composition in the

presence or in the absence of Imp PLD, we have tested for the presence of a battery of proteins known from the literature to associate with RNP complexes. As summarized in Figure VII (and revised Supplementary Figures S3A,B), we have confirmed that Imp RNP granules do not contain stress granule markers. Furthermore, we have identified a number of proteins (mainly post-transcriptional regulators) that belong to Imp RNP granules. Importantly, no difference in the recruitment of granule components was observed in the absence of the PLD domain of Imp, suggesting that the global composition of granules is not altered. We now comment this result in the main text.

	presence in Imp granules ?		contains a ...	
	Imp-wt	Imp-ΔPLD	PLD?	IDR?
Rin/GBP43	-	-	nd	nd
eIF4G	-	-	nd	nd
Rpl32	-	-	nd	nd
FMRP	-	-	nd	nd
eIF4e	+	+	no	yes
RpS6	+/-	+/-	no	no
Pur-α	+	+	no	yes
Staufen	+	+	yes	no
Me31B	+	+	no	yes
GW182	+	+	no	yes
Tral	+	+	yes	yes

Figure VII. Table summarizing the results of co-localization experiments performed with a dozen of RNP complex components.

nd stands for not determined.

This table is displayed in revised Figure S3B.

Furthermore, as some of the identified granule components contains a prion-like domain/ an intrinsically disordered region (IDR), we now also mention in our revised discussion the possibility that Imp PLD may interact with the PLDs/the IDRs of other granule components.

Also, from their previous study the authors propose that the granules contain *profilin* mRNA. However the *profilin* RNA localization in Figure S2C and S2D is actually strikingly in bright puncta that do NOT co-localize with *profilin* mRNA. While *profilin* mRNA can be detected by sensitive QRT-PCR as co-immunoprecipitating with Imp, this is only a minor fraction of the *profilin* mRNA in cells. Conversely *profilin* mRNA may be only a minor fraction of potential RNAs or other proteins associated with Imp. With these issues unaddressed, the granule properties described in the paper are lacking in both mechanism and potential meaning.

We had analyzed the distribution of *profilin* mRNA as we know from our previous study³ that i) it is directly bound by Imp, and ii) it is functionally required downstream of *imp* for axonal remodeling. To improve the quality of *profilin* mRNA detection, we have further optimized our smFISH protocol, and used higher resolution microscopy (airy scan) (Figure VIII, with the corresponding specificity control).

Figure VIII. Distribution of *profilin* (*prof*) mRNA and co-localization with Imp granules.

This panel is now displayed in revised Supplementary Figure 3D. smFISH was performed using stellaris probes. SV40 probes were used as a specificity control.

Our new results unambiguously show that a significant proportion of Imp granules (although not all) contain *profilin* mRNA, but also that *profilin* transcripts can be found outside Imp granules. This is to be expected as Imp binds to many transcripts⁴, and *profilin* molecules may not all be bound by Imp. We now comment on this point in the legend to the corresponding Supplementary Figure 3D. Importantly, our new analysis confirmed that no significant difference in the distribution of *profilin* mRNA could be observed between G080-GFP-Imp and G080-GFP-Imp-ΔPLD flies, further suggesting that granule composition is overall preserved in the absence of Imp PLD.

3- The data presented in Figure 1 are strikingly thin. It is not usual to see one circular dichroism plot with no controls or conditions for comparison.

The far UV circular dichroism analysis presented in Figure 1C reflects the conformation of amide

bonds. The spectrum obtained for Imp PLD revealed a negative peak located around 200 nm, which is a characteristic signature of a non-folded protein. Although comparing spectra obtained with increasing temperature is frequently performed to assess the response of folded domains to denaturation, this procedure is not meaningful in our case as Imp PLD domain is mainly unstructured. To provide the reader with a control condition, we now compare the CD spectrum of Imp PLD with that of Synuclein (Figure IX). We also now include a table indicating the fraction of ordered and disordered domains for both peptides.

Figure IX. Circular Dichroism spectra of Imp PLD and α -Synuclein.

This graph is displayed in revised Figure 1C.

Moreover, since the authors have purified Imp protein in hand it is puzzling that they do not examine its ability to form granules and phase separations *in vitro*. In this case they can actually know the composition, for comparison/contrast the granules imaged in cells, whose composition is not known.

We thank the reviewer for his/her suggestion. We have produced recombinant Imp proteins for analysis in *in vitro* phase separation assays. Although these proteins were produced with high quality procedures optimized in the Alberti lab, they did not behave as expected from molecularly active proteins after removal of the purification tag. In particular, they were insensitive to changes in salt concentration or to the addition of RNA molecules. Thus, although differences were observed in the maturation of Imp wt and Imp- Δ PLD condensates in these assays (See Figure VI), we don't feel confident in interpreting these experiments and would prefer not to show the results. We however now mention the technical challenges that we faced with these assays in our revised discussion.

Other more minor points:

The data in Figure 6, which are meant to show localization of GFP-Imp to MB axons is not well oriented: its very hard to see from the images what is interpreted to be 'axonal' localization and what is quantified. The text refers to some additional orientation panels in Figure S6, which is not present in this manuscript. Further orientation needs to be presented in the main figures.

We don't understand why the reviewer could not access to Figure S6 while reviewer #1 could, but apologize for this problem. To better describe what was analyzed, we have included a schematic representation of MB structures in main Figure 2 (Figure 2A), and have highlighted with dotted lines the γ axonal population in all panels of Figures 3 and 4.

Also there is a reference in error (on p.11) to Figure panels 8J-L, which are also not present, however I think the authors mean to refer to Figure 6J-L.

This has been corrected.

Reviewer #3 (Remarks to the Author):

The manuscript explores the role of a prion-like domain (PLD) of the RNA binding protein IMP in the formation of IMP positive granules (in cells and *in vivo*) and in axon migration in the *Drosophila* brain. They show that the PLD is not necessary for granule formation. The fact that PLD does not drive granule formation and is not even necessary is in line with reports that have shown that prion like domains sometimes drive granule formation and sometimes only modulate it. It suggests that either PLD or IMP in its native conformation acts as a repellent for other molecules and has a higher dynamics. However, the PLD also appears to have a positive role because whether at the Cter (as in WT IMP) or Nter, it rescues the lack of IMP in term of mushroom body recruitment, granule mobility

and axon guidance. In contrast, IMP without PLD does not.
The ms represents a lot of interesting work of high quality.

We thank the reviewer for this positive comment

However, the positive role of the PLD is understudied at the molecular level.

As described in more details below, we have now performed all technically feasible experiments to clarify this aspect.

1) The manuscript first explores the role of a prion-like domain (PLD) of the RNA binding protein IMP in the formation of IMP positive granules in *Drosophila* cells (S2 cells) and *in vivo* in the *Drosophila* mushroom body.

The authors show that this PLD is not necessary for granule formation in either system, but that PLD modulates down the number and size of the formed particles. Without it, IMP is more efficiently recruited to granules in cells and mushroom body. Conversely, when PLD is appended to human IMP1 (that does not display a PLD), granule formation is slightly impaired (at least in cells).

One line of explanation for this finding is that the presence of PLD makes the IMP molecules more mobile in and out of granules (FRAP). The recovery of full length Imp is more efficient and higher when compared to the form lacking PLD. This is further exemplified the higher background showing that the IMP molecules are overall less recruited (compare Figure 1 B versus D)

This figure should be further improved by performing more analysis.

1.1: The number of granules in mushroom body should be estimated.

In Figure 1G, GFP-IMP forms a very small number of granules, less than endogenous IMP. This should be quantified.

We agree with the reviewer that quantifying the properties of Imp granules from brain samples could be interesting. Quantification of the number and size of Imp granules from *in vivo* samples is however technically extremely challenging, as i) granule diameter is very close to the resolution limit (about 200 nm); ii) the size of MB cell bodies is small (<5 μm) and their cytoplasm is extremely thin; iii) cell bodies pile into a compact 3D structure that prevents unambiguous identification of individual cell contours. Although we have tested and optimized a large number of image analysis tools, none produced a satisfying F1 score (that incorporates the fraction of false positive and false negative detections). As our careful examination of 201Y>GFP-IMP samples (previous Figure 1G) did not reveal any obvious changes in granule number compared to endogenous Imp, we have replaced the misleading image by a more representative one (new Figure 5D).

1.2: Is there also a dynamic effect in granule formation due to the presence of PLD. Can a time course be performed? DO granules form faster in cells expressing delta PLD IMP?

This could be monitored if the expression vectors were not inducible.

Although comparing the kinetics of granule formation would be interesting, we don't think we have the resolution requested for such an analysis in cultured cells. First, Imp granules are constitutive in cells and are not induced in response to stress/external cues. Second, kinetics of granule formation can only be compared for identical levels of Imp proteins and target RNAs, two parameters that we cannot carefully control and measure in cells. Third, performing such experiments in a context where endogenous Imp is expressed might interfere with the interpretations of the results, and, downregulating endogenous *imp* has proven difficult (see Figure IV). The best-adapted system to study this aspect would be an *in vitro* reconstituted system. However, we did not pursue this option because purified Imp did not behave properly in these assays (see Figure VI).

1.3: Is the frequency of cells displaying granules the same in all conditions? This should be mentioned on top of the number of granules and their size? Is there also more cells positive for granule formation?

The fraction of cells with no detected granule in the wild-type condition is very low (<5%), so comparing this number is not very informative. To give a better idea about the range of granule

number in wild-type and Δ PLD conditions, we have now included a frequency distribution plot in Supplementary Figure 6B (see also Figure X).

Figure X. Frequency distribution of granule number upon GFP-Imp and GFP-Imp- Δ PLD expression.

The graph is displayed in revised Figure S6B.

2) Figure 4: The authors argue that the position of PLD is critical for this function as a repellent. Scrambled PLD-IMP has the same granule features as WT IMP (PLD at the C-ter). However, when PLD is at the N-ter, it acts as a form of IMP lacking PLD (larger and more granules and less background).

The reviewer argues this effect is not related to the position of the PLD. The reviewer argues that the salient feature is the absence of a sequence (any sequence) at its C-ter. Without it, IMP conformation is compromised and prevents its dynamics. Addition of PLD at the N-ter is irrelevant because the C-ter is not occupied.

- Can the authors use a method (perhaps CD) to monitor IMP conformation with or without PLD at the Cter. This is a critical experiment. It is possible that the folded WT IMP hides sticky pieces and display a different conformation. In this regard, the review took the liberty to model the structure of full length IMP, and IMP without PLD using available softwares and the conformation appears different quite different. This needs to be strengthened.

An alternative manner to evaluate a change in the IMP conformation when it lacks the PLD at Cter is to assess whether the binding affinity to mRNA as in figure 1.

We agree with the reviewer that removing the PLD of Imp could induce a change in the conformation of the rest of the protein. However, interpreting a change in RNA binding affinity by a change in protein conformation will be difficult, as low complexity domains were shown in different contexts to contribute to RNA binding^{5,6}. We find that CRISPR flies expressing Δ PLD Imp are viable and fertile, in contrast to imp loss-of-function flies. This demonstrates that the integrity of the protein is largely preserved in the absence of the PLD and that the PLD is not required for the essential functions of Imp. We now emphasize this point more strongly in the revised manuscript.

At this point, we cannot exclude that the lack of Imp PLD induces a subtle change in conformation, but such a change is unlikely to be detected using Circular Dichroism. Identifying such subtle conformational changes requires solving the structure of wild-type and Δ PLD protein. This represents an entire new project that we feel is beyond the scope of this manuscript, especially as our attempts have already shown that purified Imp protein is extremely difficult to work with (J. Vijayakumar, S. Alberti, L. Bousset, A. Ramos, unpublished work). Thus, although we cannot formally exclude that Imp undergoes subtle changes in protein conformation in the absence of the PLD, we think that this is very unlikely. To address the concern raised by the reviewer further, we now discuss this possibility in the revised discussion of our manuscript.

Furthermore, identifying the interactome of full length IMP and IMP lacking PLD reveal possible important differences.

To address whether the composition of Imp granules was altered in the absence of Imp PLD, we have analyzed the distribution of a dozen of granule-associated markers. As shown in Figure VII, and as now reported in the manuscript (Supplementary Figure S3A-C), all markers present in wild-type Imp granules were also present in Imp- Δ PLD granules, arguing against a general disorganization of Imp granules. At this point, we cannot exclude the possibility that one/few selected partners are differentially recruited. Whether such differences could be attributed to changes in protein

conformation is however unclear.

3) In this context, the experiment presented in Figure 4 should be completed by 3 additional tests.

3.1: IMP C-ter should be occupied by an irrelevant sequence, very different from PLD. Afterall, scrambling PLD that is so rich in Q will end up in rather similar sequences. What about a totally different sequence of the same length? Will this behave a WT IMP or delta PLD IMP?

To address the point raised by the reviewer, we generated a Imp- Δ PLD-GFPstart variant in which Imp PLD sequence was replaced by an irrelevant sequence of similar length (first part of GFP). As shown in Figure XI, the number of granules formed upon expression of Imp- Δ PLD-GFPstart was still significantly higher than that of Imp-wt, suggesting that adding a non-relevant C-terminal sequence cannot compensate for the lack of the PLD, and thus that the PLD encodes specific information. This result is now displayed in the revised Supplementary Figure 7G and commented in the main text.

Figure XI. Distribution of Imp granule number upon expression of Imp- Δ PLD-GFPstart.

In the Imp- Δ PLD-GFPstart construct, the PLD sequence has been replaced by a fragment of similar length from the GFP sequence. Note that both Imp- Δ PLD and Imp- Δ PLD-GFPstart fusions show increased granule number.

This graph is now displayed in revised Supplementary Figure 7G.

3.2: Can this chimera (or the one with Scrambled PLD) be appended with real PLD at the N-ter?

If the conformation is rescued with addition of any sequence at the C-ter, perhaps a role for PLD in granule formation can be discovered in line with its role in axons independent of granule dynamics.

As the results described in 3.1 (Figure XI) argue against the hypothesis that any sequence can compensate for the lack of Imp PLD, we have not performed the proposed experiment.

3.3: In this context, the human IMP1 should be appended by scrambled dros PLD, not the WT form. Again, addition of the C-ter might fold the protein differently and be enough to affect granule formation.

As suggested by the reviewer, we have generated hIMP1-drosPLD chimeric proteins using scrambled versions of the PLD (scr2 and scr4). As shown in Figure XII and in the revised Supplementary Figure 7F, these chimeric proteins behave similarly to the original hIMP1-drosPLD, further highlighting that the primary sequence of the PLD is not important to restrict granule number.

Figure XII. Distribution of Imp granule number upon expression of hIMP1-drosPLD-scr constructs.

Note that hIMP1-drosPLD-scr constructs behave similarly to hIMP1-drosPLD ones.

This graph is displayed in revised Supplementary Figure 7F.

4) The authors monitor IMP recruitment to the mushroom body and show that without PLD, IMP is not present in the mushroom body.

The reviewer is not an expert but he wonders how can Figure 1 J be achieved if delta PLD-IMP cannot reach the mushroom bodies.

Mushroom Bodies (MBs) are composed of hundreds of γ neurons, each of them containing a cell body and axonal extensions. While Imp- Δ PLD expressed by MB γ neurons accumulate in cell bodies similarly to wild-type Imp (see Supplementary Figure 2A and Figure 5), they are not transported to their axons properly (Figure 3). To clarify this point, we have inserted a schematic representation of MBs and their constituent neurons in main Figure 2A.

5) Interestingly, N-ter PLD is recruited to the MB, and so is hIMP1-dros PLD. This suggests a positive role for PLD on top of occupying the C-ter. Is this related to the conformation? (see above).

We cannot exclude this possibility, but as the reviewer pointed out, Imp PLD is instructive for axonal transport, whether located at the N-terminus of the *Drosophila* protein, or at the C-terminus of the hIMP1 protein. How it would in both cases induce a similar change in conformation is unclear to us.

6) Figure 7: The granule dynamics is virtually similar (with the speed of delta PLD-IMP + positive granules slightly higher than full length IMP). What does it mean?

As shown in various living systems⁷⁻⁹, subcellular targeting of cargo molecules is achieved by small biases in the properties of the bidirectional transport of motile granules. Such biases in cargo velocities are most frequently due to differences in the number of recruited molecular motors, or in their activity. As noted by the referee, a small increase in the velocity of retrograde Imp granules is observed in the Imp- Δ PLD context. Such an increase might reflect a differential recruitment of + end and – end directed molecular motors, or an imbalance in their relative activity. The major impact observed upon PLD removal is a decrease in the number of motile granules, and we think this reduction is the main driver of the observed axon localization defects. To better illustrate the observed changes in the velocity of retrograde granules we have included a frequency distribution of retrograde run velocities (Supplementary Figure 4C).

Would Nter PLD-IMP act as WT-IMP or delta PLD-IMP as suggested by experiments presented in Figure 8?

Although this experiment would be interesting, it would rely on the analysis of granules formed upon over-expression of GFP-fusion proteins (Gal4/UAS system). As we and others^{3,10,11} have observed, granules formed in these conditions are larger than the endogenous ones and may not entirely reflect the properties of endogenous ones.

7) Sadly, the positive role of PLD in recruitment to MB and axon morphology is understudied and is preliminary. Has any interactors of PLD be determined perhaps by mass spec that could explain this role? If the delta PLD has a different conformation, the mass spec experiment will need to be carefully controlled to pick up specific interactors (see also point above).

We have performed an IP Mass-Spectrometry analysis from brain extracts and did not identify any differential enrichment of microtubule-dependent motors or regulatory proteins when quantitatively comparing Imp and Imp- Δ PLD pull-downs. This is not unexpected as i) motors are sticky proteins/complexes that produce significant background; ii) in most cases motor-cargo associations are transient and not very stable. To properly address this question, one would probably need to purify granules rather than immuno-precipitating Imp proteins, which is currently not technically possible given the small size of the granules and the quantity of material required.

References

- 1- Chao, J. A. *et al.* ZBP1 recognition of beta-actin zipcode induces RNA looping. *Genes Dev* **24**, 148-158, doi:10.1101/gad.1862910 (2010).
- 2- Song, T. *et al.* Specific interaction of KIF11 with ZBP1 regulates the transport of beta-actin mRNA and cell motility. *J Cell Sci* **128**, 1001-1010, doi:10.1242/jcs.161679 (2015).

- 3- Medioni, C., Ramialison, M., Ephrussi, A. & Besse, F. Imp promotes axonal remodeling by regulating profilin mRNA during brain development. *Curr Biol* **24**, 793-800, doi:10.1016/j.cub.2014.02.038 (2014).
- 4- Hansen, H. T. *et al.* Drosophila Imp iCLIP identifies an RNA assemblage coordinating F-actin formation. *Genome Biol* **16**, 123, doi:10.1186/s13059-015-0687-0 (2015).
- 5- Castello, A. *et al.* Comprehensive Identification of RNA-Binding Domains in Human Cells. *Mol Cell* **63**, 696-710, doi:10.1016/j.molcel.2016.06.029 (2016).
- 6- Lin, Y., Protter, D. S., Rosen, M. K. & Parker, R. Formation and Maturation of Phase-Separated Liquid Droplets by RNA-Binding Proteins. *Mol Cell* **60**, 208-219, doi:10.1016/j.molcel.2015.08.018 (2015).
- 7- Hancock, W. O. Bidirectional cargo transport: moving beyond tug of war. *Nat Rev Mol Cell Biol* **15**, 615-628, doi:10.1038/nrm3853 (2014).
- 8- Jolly, A. L. & Gelfand, V. I. Bidirectional intracellular transport: utility and mechanism. *Biochem Soc Trans* **39**, 1126-1130, doi:10.1042/BST0391126 (2011).
- 9- Turner-Bridger, B. *et al.* Single-molecule analysis of endogenous beta-actin mRNA trafficking reveals a mechanism for compartmentalized mRNA localization in axons. *Proc Natl Acad Sci U S A* **115**, E9697-E9706, doi:10.1073/pnas.1806189115 (2018).
- 10- De Graeve, F. & Besse, F. Neuronal RNP granules: from physiological to pathological assemblies. *Biol Chem* **399**, 623-635, doi:10.1515/hsz-2018-0141 (2018).
- 11- Mazroui, R. *et al.* Trapping of messenger RNA by Fragile X Mental Retardation protein into cytoplasmic granules induces translation repression. *Hum Mol Genet* **11**, 3007-3017 (2002).

Reviewers' comments:

Reviewer #1 (Remarks to the Author):

The authors have certainly improved the manuscript with additional experiments, controls, and figures. This reviewer certainly believes that the reorganization of the text, such that it leads with ascertaining the biological function of Imp's PLD domain, was wise and improves the flow of the manuscript substantially. The PLD domain is important for Imp granule axon localization, correct growth, and axonal transport. Furthermore, the data herein support a regulatory role of the PLD domain in granule size and dynamics. While the connection of granule dynamics to biological function is still not clear, it sets the stage for future studies. The authors did attempt to execute in vitro phase sep. experiments of Imp and Imp deletion constructs, but these data are too preliminary and should not be included in the manuscript, as the authors suggest. Further work will be needed to determine the molecular basis for PLD function. Overall, the manuscript is well-written.

Minor corrections are suggested for suitable publication in Nature Communications.

- What is the effect of N-term PLD placement with respect to Imp granule motility (anterograde vs retrograde transport)? I ask, only because there is no effect on defective polarized growth with N-term PLD compared to WT, but there is a significant change in granule dynamics (Fig 9). Perhaps this may strengthen the connection of granule size/dynamics to Imp granule function.

- For sake of data transparency, could Figures 3C,F,I and Figures 4F,G and Figures 7D,H and Figure 9D also include dot plots to see data spread?

- Text corrections:

bottom of pg. 5: "we generated flies that express proteins lacking the PLD (Imp- Δ PLD) USING THE CRISPR/Cas9 TECHNOLOGY."

bottom of pg. 10: "grafting scrambled PLDs ONTO hIMP1"

on top of pg. 14: "Such a grammar is not APPLICABLE TO Imp PLD..."

Reviewer #2 (Remarks to the Author):

I appreciate the additions and changes that the authors have made which indeed have improved the manuscript. However the observations are still not quite coming together into a coherent story. My main criticism at this point is the major gap in understanding how the in vivo localization and functional data might possibly relate the role of the PLD in the granule properties of Imp. This might be addressable.

Given the hypothesized importance of RNPs in axons, the mechanism for the striking axonal localization defect of the PLD domain mutant deserves more attention. The authors present live imaging data of granule motility in MB axons but this makes more questions than answers:

1) is the axonally localized GFP-Imp in Figure 4 generally in granules or diffuse/cytoplasmic? Are FRAP studies possible in this preparation?

2) is granule formation required for axonal localization? The authors have RNA-binding mutant that does not form granules. They should present axon localization data for this mutant.

I frankly do not know what to make of the modest changes in granule motility in axons that were measured by live imaging in Figure 6. The authors interpret that the PLD domain may participate in the linkage of the granules to kinesin and dynein motors. But an alternate interpretation of this data could also be made: Figure 6 shows that the granules are in general moving quite well in the deltaPLD mutants. This defect seems modest compared to the axon localization defect. Since the FRAP data suggest that Imp-deltaPLD protein exchanges less well between the granules and the

cytoplasm, I am wondering if the PLD helps Imp localize to axoplasm by enabling dissociation from the granules. This model has some appeal – granule properties could be generally useful for a transport mechanism: the granule can be a substrate for transport, but dynamics out of the granule could serve to translate transport by a motor into localization throughout a large area of a cell with a long axon. The paper as currently written does not really present a coherent model. Are their data consistent with something like this?

Reviewer #3 (Remarks to the Author):

The authors have performed a great job in revising their, adding new data and turning it around as suggested by reviewer 2, thus putting emphasis on the in vivo data that show a role for PLD in Imp.

The conclusion of their ms is that the Imp PLD is important for granule transport in the axon. This is not correlated to granule formation but correlates well with their dynamics.

Of interesting note: Granules do form in the absence of PLD. This is a opposite result to the current view on PLD. It is therefore important that the authors emphasize this result and raise awareness that low complexity sequences is certainly not the all story behind granule formation.

The authors have specifically addressed most of my comments by performing nearly all requested experiments, including replacing the PLD by irrelevant sequences. There is definitely a role for PLD (wt or scrambled) beyond a change in conformation. The fact that flies bearing a version of IMP without PLD are viable and fertile suggest indeed that the protein conformation is not grossly modified.

I am therefore happy with the state of the revised ms including its new presentation, and I am in favor of its publication in Nat Comm.

Reviewer #1 (Remarks to the Author):

The authors have certainly improved the manuscript with additional experiments, controls, and figures. This reviewer certainly believes that the reorganization of the text, such that it leads with ascertaining the biological function of Imp's PLD domain, was wise and improves the flow of the manuscript substantially. The PLD domain is important for Imp granule axon localization, correct growth, and axonal transport. Furthermore, the data herein support a regulatory role of the PLD domain in granule size and dynamics. While the connection of granule dynamics to biological function is still not clear, it sets the stage for future studies. The authors did attempt to execute *in vitro* phase sep. experiments of Imp and Imp deletion constructs, but these data are too preliminary and should not be included in the manuscript, as the authors suggest. Further work will be needed to determine the molecular basis for PLD function. Overall, the manuscript is well-written.

Minor corrections are suggested for suitable publication in Nature Communications.

- What is the effect of N-term PLD placement with respect to Imp granule motility (anterograde vs retrograde transport)? I ask, only because there is no effect on defective polarized growth with N-term PLD compared to WT, but there is a significant change in granule dynamics (Fig 9). Perhaps this may strengthen the connection of granule size/dynamics to Imp granule function.

The point raised by the reviewer is interesting, but cannot be addressed with current tools. For our *in vivo* analysis of Imp granule motility, indeed, we have monitored the behavior of endogenously expressed proteins using flies engineered with the CRISPR/Cas9 technology. Imaging endogenous granules is very important, as previous reports showed that overexpressed proteins often accumulate into granules with abnormal kinetics. This is however not currently possible for the Imp-Nter-PLD variant expressed from an artificial promoter.

- For sake of data transparency, could Figures 3C,F,I and Figures 4F,G and Figures 7D,H and Figure 9D also include dot plots to see data spread?

We have modified all box plots so that the distribution of individual data points is now displayed on the graph.

- Text corrections:

bottom of pg. 5: "we generated flies that express proteins lacking the PLD (Imp- Δ PLD) USING THE CRISPR/Cas9 TECHNOLOGY."

bottom of pg. 10: "grafting scrambled PLDs ONTO hIMP1"

on top of pg. 14: "Such a grammar is not APPLICABLE TO Imp PLD..."

All suggested changes have been made.

Reviewer #2 (Remarks to the Author):

I appreciate the additions and changes that the authors have made which indeed have improved the manuscript. However the observations are still not quite coming together into a coherent story. My main criticism at this point is the major gap in understanding how the *in vivo* localization and functional data might possibly relate the role of the PLD in the granule properties of Imp. This might be addressable.

We thank the reviewer for his/her positive comments and further suggestions. We however would like to emphasize that our results have suggested that the localization defects observed in the absence of Imp PLD are not linked to changes in granule properties. Although surprising, this raises the interesting possibility of a direct function for a PLD in transport. We nox better highlight this idea in our revised discussion.

Given the hypothesized importance of RNPs in axons, the mechanism for the striking axonal localization defect of the PLD domain mutant deserves more attention. The authors present live imaging data of granule motility in MB axons but this makes more questions than answers:

1) is the axonally localized GFP-Imp in Figure 4 generally in granules or diffuse/cytoplasmic?

To address the reviewer's question, we have performed high-resolution confocal imaging of Imp distribution in axons. This revealed first that Imp localizes as puncta in axons and second that the

overall density of axonal Imp punctae is reduced in the absence of Imp PLD (see below). These new results are presented in the revised Figure 3.

Magnified view of the distal part of medial MB γ lobes from G080-GFP-Imp (left) and G080-GFP-Imp- Δ PLD (right) adult brains. GFP signals are represented using

the “Fire” look-up table of ImageJ. Scale bar: 3 μ m.

Are FRAP studies possible in this preparation?

Although FRAP studies on axonally-localized granules would be interesting, this is technically not possible as axons are localized more deeply in the brain than cell bodies, preventing high-quality and high-resolution imaging of distinct granules.

2) is granule formation required for axonal localization? The authors have RNA-binding mutant that does not form granules. They should present axon localization data for this mutant.

As suggested by the reviewer, we have quantitatively analyzed the axonal localization of the Imp-KH1-4DD variant that cannot assemble into visible cytoplasmic granules. As shown below, KH1-4DD variants localized less efficiently than wild-type Imp to axons, again consistent with the idea that Imp proteins travel to axons as granules. This new result is now shown in the revised Supplementary Fig. 4d and mentioned in the revised discussion.

Distributions of normalized GFP signal intensities in distal axons (Min to max box plots).
***, $P < 0.001$ (Mann-Whitney test).

I frankly do not know what to make of the modest changes in granule motility in axons that were measured by live imaging in Figure 6. The authors interpret that the PLD domain may participate in the linkage of the granules to kinesin and dynein motors. But an alternate interpretation of this data could also be made: Figure 6 shows that the granules are in general moving quite well in the deltaPLD mutants. This defect seems modest compared to the axon localization defect. Since the FRAP data suggest that Imp-deltaPLD protein exchanges less well between the granules and the cytoplasm, I am wondering if the PLD helps Imp localize to axoplasm by enabling dissociation from the granules. This model has some appeal – granule properties could be generally useful for a transport mechanism: the granule can be a substrate for transport, but dynamics out of the granule could serve to translate transport by a motor into localization throughout a large area of a cell with a long axon. The paper as currently written does not really present a coherent model. Are their data consistent with something like this?

The reviewer proposes an interesting model in which the overall Imp signal observed *in vivo* in MB γ axons would largely correspond to molecules that have dissociated from granules after transport. Based on this hypothesis and on the fact that Imp- Δ PLD molecules assemble more efficiently into granules, he/she proposes that the strong decrease in axonal signal observed in the absence of Imp PLD might actually reflect a lack of dissociation from axonal granules. Although appealing, this model is not consistent with the new high-resolution imaging of Imp axonal signal that we now present in our revised manuscript. As shown in revised Figure 3, Imp proteins, indeed, accumulate in axons as granular structures rather than diffusely, and a decreased density of Imp puncta is observed in the Imp- Δ PLD context. In addition, this model does not explain why the Imp-Nter-PLD variant, that assembles with increased efficiency into granules, exhibit a similar axonal enrichment than wild-type Imp.

As pointed out by the reviewer, the motility of axonal Imp granules is not completely abolished in the absence of Imp PLD. A significant decrease (>1.5 fold) in the number of motile granules is however

observed, that could on a physiological time scale lead to a significant decrease in axonal accumulation. As we cannot exclude that a fraction of Imp molecules travels independently of detectable granules due to the limitations of confocal microscopy live-imaging analysis, we now discuss this possibility in our revised discussion.

Reviewer #3 (Remarks to the Author):

The authors have performed a great job in revising their, adding new data and turning it around as suggested by reviewer 2, thus putting emphasis on the in vivo data that show a role for PLD in Imp. The conclusion of their ms is that the Imp PLD is important for granule transport in the axon. This is not correlated to granule formation but correlates well with their dynamics.

Of interesting note: Granules do form in the absence of PLD. This is a opposite result to the current view on PLD. It is therefore important that the authors emphasize this result and raise awareness that low complexity sequences is certainly not the all story behind granule formation.

The authors have specifically addressed most of my comments by performing nearly all requested experiments, including replacing the PLD by irrelevant sequences. There is definitely a role for PLD (wt or scrambled) beyond a change in conformation. The fact that flies bearing a version of IMP without PLD are viable and fertile suggest indeed that the protein conformation is not grossly modified.

I am therefore happy with the state of the revised ms including its new presentation, and I am in favor of its publication in Nat Comm.

We thank the reviewer for his/her very positive comments.

REVIEWERS' COMMENTS:

Reviewer #1 (Remarks to the Author):

I appreciate the authors' response to my questions and concerns. I find the manuscript ready for publication in Nature Communications. The story is timely. The concept that the PLD modulates granule assembly and dynamics rather than promotes granule formation will be of significant interest to others studying PLDs and how they regulate biomolecular condensates.

Reviewer #2 (Remarks to the Author):

I appreciate the additions provided by the authors and feel a weariness with the back-and-forth that I suspect everyone shares. However, as I stated in the previous review, the analysis of the RNA binding mutant, which should not form granules, is an important control. The authors included this in the revision in a quantification buried in a supplemental figure but did not show any images. This is a missed opportunity to see a valuable control for the faint puncta shown in the lobes (in Figure 3e and e') that are interpreted to be granules. I appreciate that these objects are quite hard to image and appear close to background. All the more reason to include controls side-by-side with a GFP-Imp-RNA-binding domain mutant construct which is reported to not form granules. Since the relationship of the granule forming properties to the localization is the major focus of this paper, the readers should be able to see and evaluate this control for comparison to the other localization data.

Point-by-point responses to the referees:

Reviewer #1:

I appreciate the authors' response to my questions and concerns. I find the manuscript ready for publication in Nature Communications. The story is timely. The concept that the PLD modulates granule assembly and dynamics rather than promotes granule formation will be of significant interest to others studying PLDs and how they regulate biomolecular condensates.

We thank the reviewer for his/her positive comments.

Reviewer #2:

I appreciate the additions provided by the authors and feel a weariness with the back-and-forth that I suspect everyone shares. However, as I stated in the previous review, the analysis of the RNA binding mutant, which should not form granules, is an important control. The authors included this in the revision in a quantification buried in a supplemental figure but did not show any images. This is a missed opportunity to see a valuable control for the faint puncta shown in the lobes (in Figure 3e and e') that are interpreted to be granules. I appreciate that these objects are quite hard to image and appear close to background. All the more reason to include controls side-by-side with a GFP-Imp-RNA-binding domain mutant construct which is reported to not form granules. Since the relationship of the granule forming properties to the localization is the major focus of this paper, the readers should be able to see and evaluate this control for comparison to the other localization data.

We have now included in Figures 3 d and e the control images requested by Reviewer #2. These high-resolution images highlight that GFP-Imp wild-type (Fig. 3d), in contrast to GFP-Imp-KH1-4DD (Fig. 3e), localize as granules in the axons of MB γ neurons.